# Disentangling Adversarial Robustness in Directions of the Data Manifold

## Abstract

Using generative models (GAN or VAE) to craft adversarial examples, i.e. generative adversarial examples, has received increasing attention in recent years. Previous studies showed that the generative adversarial examples work differently compared to that of the regular adversarial examples in many aspects, such as attack rates, perceptibility, and generalization. But the reasons causing the differences between regular and generative adversarial examples are unclear. In this work, we study the theoretical properties of the attacking mechanisms of the two kinds of adversarial examples in the Gaussian mixture model. We prove that adversarial robustness can be disentangled in directions of the data manifold. Specifically, we find that: 1. Regular adversarial examples attack in directions of small variance of the data manifold, while generative adversarial examples attack in directions of large variance. 2. Standard adversarial training increases model robustness by extending the data manifold boundary in directions of small variance, while on the contrary, adversarial training with generative adversarial examples increases model robustness by extending the data manifold boundary directions of large variance. In experiments, we demonstrate that these phenomena also exist on real datasets. Finally, we study the robustness trade-off between generative and regular adversarial examples. We show that the conflict between regular and generative adversarial examples is much smaller than the conflict between regular adversarial examples of different norms.

## 1 Introduction

In recent years, deep neural networks (DNNs) (Krizhevsky et al. (2012); Hochreiter and Schmidhuber (1997)) have become popular and successful in many machine learning tasks. They have been used in different problems with great success. But DNNs are shown to be vulnerable to adversarial examples (Szegedy et al. (2013); Goodfellow et al. (2014a)). A well-trained model can be easily attacked by adding a small perturbation to the image. An effective way to solve this issue is to train the robust model using training data augmented with adversarial examples, i.e. adversarial training.

With the growing success of generative models, researchers have tried to use generative adversarial networks (GAN) (Goodfellow et al. (2014b)) and variational autoencoder (VAE) (Kingma and Welling (2013)) to generate adversarial examples (Xiao et al. (2018); Zhao et al. (2017); Song et al. (2018a); Kos et al. (2018); Song et al. (2018b)) to fool the classification model with great success. They found that standard adversarial training cannot defend these new attacks. Unlike the regular adversarial examples, these new adversarial examples are perceptible by humans but they preserve the semantic information of the original data. A good DNN should be robust to such semantic attacks. Since the GAN and VAE are approximations of the true data distribution, these adversarial examples will stay in the data manifold. Hence they are called **on-manifold** adversarial examples by (Stutz et al. (2019)). On the other hand, experimental evidences support that regular adversarial examples leave the data manifold (Song et al. (2017)). We call the regular adversarial examples as **off-manifold** adversarial examples.

The concepts of on-manifold and off-manifold adversarial examples are important. Because they can help us to understand the issue of conflict between adversarial robustness and generalization (Stutz et al. (2019); Raghunathan et al. (2019)), which is still an open problem. In this paper, we study the attacking mechanisms of these two types of examples, as well as the corresponding adversarial

training methods. This study, as far as we know, has not been done before. Specifically, we consider a generative attack method that adds a small perturbation in the latent space of the generative models. Since standard adversarial training cannot defend this attack, we consider the training methods that use training data augmented with these on-manifold adversarial examples, which we call latent space adversarial training. Then we compare it to standard adversarial training (training with off-manifold adversarial examples).

**Contributions:** We study the theoretical properties of latent space adversarial training and standard adversarial training in the Gaussian mixture model with a linear generator. We give the excess risk analysis and saddle point analysis in this model. Based on this case study, we claim that:

- Regular adversarial examples attack in directions of small variance of the data manifold and leave the data manifold.

- Standard adversarial training increases the model robustness by amplifying the small variance. Hence, it extends the boundary of the data manifold in directions of small variance.

- Generative adversarial examples attack in directions of large variance of the data manifold and stay in the data manifold.

- Latent space adversarial training increases the model robustness by amplifying the large variance. Hence, it extends the boundary of the data manifold in directions of large variance.

We provide experiments on MNIST and CIFAR-10 and show that the above phenomena also exist in real datasets. It gives us a new perspective to understand the behavior of on-manifold and off-manifold adversarial examples. Finally, we study the robustness trade-off between generative and regular adversarial examples. On MNIST, robustness trade-off is unavoidable, but the conflict between generative adversarial examples and regular adversarial examples are much smaller than the conflict between regular adversarial examples of different norms. On CIFAR-10, there is nearly no robustness trade-off between generative and regular adversarial examples.

## 2 RELATED WORK

Our work is related to attack and defense methods. Specifically, we care about attacks and defenses with generative models.

**Attack** Adversarial examples for deep neural networks were first intruduced in (Szegedy et al. (2013)). However, adversarial machine learning or robust machine learning has been studied for a long time (Biggio and Roli (2018)). In the setting of **white box attack** (Kurakin et al. (2016); Papernot et al. (2016); Moosavi-Dezfooli et al. (2016); Carlini and Wagner (2017)), the attackers have fully access to the model (weights, gradients, etc.). In **black box attack** (Chen et al. (2017); Su et al. (2019); Ilyas et al. (2018)), the attackers have limited access to the model. First order optimization methods, which use the gradient information to craft adversarial examples, such as PGD (Madry et al. (2017)), are widely used for white box attack. Zeroth-order optimization methods (Chen et al. (2017)) are used in black box setting. Li et al. (2019) improved the query efficiency in black-box attack. HopSkipJumpAttack (Chen et al. (2020)) is another query-efficient attack method.

**Generative adversarial examples** Recently, generative models have been used to craft adversarial examples (Xiao et al. (2018); Song et al. (2018b); Kos et al. (2018); Schott et al. (2018)). The adversarial examples are more natural (Zhao et al. (2017)). These adversarial examples lie in the data manifold, and they are called on-manifold adversarial examples.

**Defense** Training algorithms against adversarial attacks can be subdivided into the following categories. **Adversarial training:** The training data is augmented with adversarial examples to make the models more robust (Madry et al. (2017); Szegedy et al. (2013); Tramèr et al. (2017)). **Preprocessing:** Inputs or hidden layers are quantized, projected onto different sets or other preprocessing methods (Buckman et al. (2018); (Guo et al. (2017); Kabilan et al. (2018)). **Stochasticity:** Inputs or hidden activations are randomized (Prakash et al. (2018); Dhillon et al. (2018); Xie et al. (2017)). However, some of them are shown to be useless defenses given by obfuscated gradients (Athalye et al. (2018)). Adaptive attack (Tramer et al. (2020)) is used for evaluating defenses to adversarial examples.

**Defense with generative model** Using generative models to design defense algorithms have been studied extensively. Using GAN, we can project the adversarial examples back to the data manifold

(Jalal et al. (2017); Samangouei et al. (2018)). VAE is also used to train robust model (Schott et al. (2018)).

## 3 PROBLEM DESCRIPTION

**Original space adversarial training:**  Consider the classification problem of training a classifer $f_\theta$ to map the data points $x \in \mathcal{X} \subset \mathbb{R}^d$ to the labels $y \in \mathcal{Y}$, where $\mathcal{X}$ and $\mathcal{Y}$ are the input data space and the label space. The classifier $f_\theta$ is parameterized by $\theta$. We assume that the data pairs $(x, y)$ are sampled from the distribution $P(X, Y)$ over $\mathcal{X} \times \mathcal{Y}$. Standard training is to find the solution of $\min_\theta \mathbb{E}_{(x,y) \sim P} \ell(f_\theta(x), y)$, where $\ell(\cdot, \cdot)$ is the loss function. The goal of dversarial training is to solve the minimax problem

$$\min_\theta \mathbb{E}_{(x,y) \sim P} \max_{\|x - x'\| \leq \varepsilon} \ell(f_\theta(x'), y), \tag{1}$$

where $\varepsilon$ is the threshold of perturbation. Here we can use $\ell_1$, $\ell_2$ or $\ell_\infty$-norm (Madry et al. (2017)). The inner maximization problem is to find the adversarial examples $x'$ to attack the given classifier $f_\theta$. The outer minimization problem is to train the classifier to defend the given adversarial examples $x'$. we refer to these attacks as regular attacks. We refer to these minimax problems as standard adversarial training or original space adversarial training.

**Latent space adversarial training:**  We assume that the data lie in a low dimensional manifold of $\mathbb{R}^d$. Furthermore, we assume the true distribution $\mathcal{D}$ is a pushforward from a prior Guassian distribution $z \sim \mathcal{N}(0, I)$ using $G(z)$, where $G : \mathcal{Z} \to \mathcal{X}$ is a mapping from the latent space $\mathcal{Z}$ to the original space $\mathcal{X}$. This is a basic assumption of GAN or VAE. Let $I : \mathcal{X} \to \mathcal{Z}$ be the inverse mapping of $G(z)$. The goal of latent space adversarial training is to solve the following minimax problem

$$\min_\theta \mathbb{E}_{(x,y) \sim P} \max_{\|z' - I(x)\| \leq \varepsilon} \ell(f_\theta(G(z')), y). \tag{2}$$

Unlike the regular attacks, the distance between the original examples and adversarial examples can be large. To preserve the label of the data, we use the conditional generative models (e.g. C-GAN (Mirza and Osindero (2014)) and C-VAE (Sohn et al. (2015))), i.e. the generator $G_y(z)$ and inverse mapping $I_y(x)$ are conditioned on the label $y$, for adversarial training. We refer to these attacks as generative attack, and these adversarial training as latent space adversarial training.

**Regular attack algorithms**  Two widely used gradient-based attack algorithms for the inner maximization problem in equation (1) are **fast gradient sign method (FGSM)** (Goodfellow et al. (2014a)) and **projected gradient descend (PGD)** (Madry et al. (2017)). Using FGSM, the adversarial examples are calculated by

$$x' = x + \varepsilon \text{sgn}(\nabla_x \ell(f_\theta(x), y)),$$

where $\nabla_x$ denotes the gradient with respect to $x$. PGD attempts to find a near optimal adversarial example for the inner maximization problem (1) in multiple steps. In the $t^{th}$ step,

$$x^{t+1} = \Pi_{x+S}[x^t + \alpha \nabla_x \ell(f_\theta(x^t), y) / \|\nabla_x \ell(f_\theta(x^t), y)\|],$$

where $\alpha$ is the step size, $\Pi_{x+S}[\cdot]$ is the projection operator to project the given vector to the constraint $x + S = \{x' | \|x - x'\| \leq \varepsilon\}$.

In the whole paper, we refer to these as FGSM-attack and PGD-attack, and the corresponding original space adversarial training as FGSM-adv and PGD-adv. FGSM-attack is a weak attack and PGD-attack is a stronger attack. In section 5, we use them to show that a strong original space adversarial training, PGD-adv, does not work well against a weak attack, FGSM-attack in the latent space. Conversely, latent space adversarial training cannot defend a simple FGSM-attack in the original space.

**Generative attack algorithm**  In our experiments, we use FGSM in the latent space for the inner maximization problem in equation (2)

$$z' = I(x) + \varepsilon \text{sgn}(\nabla_z \ell(f_\theta(G(z)), y)).$$

Because of the mode collapse issue of GAN (Salimans et al. (2016); Gulrajani et al. (2017)), adding a small perturbation in the latent space of GAN may output the same images. Thus we use VAE in our experiments. we refer to this generative attack and latent space adversarial training as VAE-attack and VAE-adv.

# 4 THEORETICAL ANALYSIS

In this section, we study the difference between adversarial training in the latent space and in the original space. We study the simple binary classification setting proposed by Ilyas et al. (2019). The main reason for using this model is that we can find the optimal closed-form solution, which gives us insights to understand adversarial training. For a more complex model, we can only solve it numerically. We leave all the proofs of the lemmas and theorems in appendix A.

## 4.1 THEORETICAL MODEL SETUP

**Gaussian mixture (GM) model** Assume that data points $(x, y)$ are sampled according to $y \sim \{-1, 1\}$ unifomly and $x \sim \mathcal{N}(y\boldsymbol{\mu}_*, \boldsymbol{\Sigma}_*)$, where $\boldsymbol{\mu}_*$ and $\boldsymbol{\Sigma}_*$ denote the true mean and covariance matrix of the data distribution. For the data in the class $y = -1$, we replace $x$ by $-x$, then we can view the whole dataset as sampled from $\mathcal{D} = \mathcal{N}(\boldsymbol{\mu}_*, \boldsymbol{\Sigma}_*)$.

**Classifier** The goal of standard training is to learn the parameters $\boldsymbol{\Theta} = (\boldsymbol{\mu}, \boldsymbol{\Sigma})$ such that

$$\boldsymbol{\Theta} = \arg\min_{\boldsymbol{\mu}, \boldsymbol{\Sigma}} \mathcal{L}(\boldsymbol{\mu}, \boldsymbol{\Sigma}) = \arg\min_{\boldsymbol{\mu}, \boldsymbol{\Sigma}} \mathbb{E}_{x \sim \mathcal{D}}[\ell(x; \boldsymbol{\mu}, \boldsymbol{\Sigma})], \tag{3}$$

where $\ell(\cdot)$ represents the negative log-likelihood function. The goal of adversarial training is to find

$$\boldsymbol{\Theta_r} = \arg\min_{\boldsymbol{\mu}, \boldsymbol{\Sigma}} \mathcal{L}_r(\boldsymbol{\mu}, \boldsymbol{\Sigma}) = \arg\min_{\boldsymbol{\mu}, \boldsymbol{\Sigma}} \mathbb{E}_{x \sim \mathcal{D}}[\max_{\|x - x'\| \leq \varepsilon} \ell(x'; \boldsymbol{\mu}, \boldsymbol{\Sigma})]. \tag{4}$$

We use $\mathcal{L}$ and $\mathcal{L}_r$ to denote the standard loss and adversarial loss. After training, we classify a new data point $x$ to the class $\mathrm{sgn}(\boldsymbol{\mu}^T \boldsymbol{\Sigma}^{-1} x)$.

**Generative model** In our theoretical study, we use a linear generative model, that is, probabilistic principle components analysis (P-PCA) (Tipping and Bishop (1999)). P-PCA can be viewed as linear GAN (Feizi et al. (2017); Feizi et al. (2020)) and linear VAE (Dai et al. (2017)).

Given dataset $\{x_i\}_{i=1}^n \subset \mathbb{R}^d$, let $\boldsymbol{\mu}$ and $\boldsymbol{S}$ be the sample mean and sample covariance matrix. The eigenvalue decomposition of $\boldsymbol{S}$ is $\boldsymbol{S} = \boldsymbol{U}\boldsymbol{\Lambda}\boldsymbol{U}^T$, then using the first $q$ eigenvactors, we can project the data to a low dimensional space. P-PCA is to assume that the data are generated by

$$x = \boldsymbol{W}z + \boldsymbol{\mu} + \boldsymbol{\epsilon} \quad \text{where} \quad z \sim \mathcal{N}(0, I), \ \boldsymbol{\epsilon} \sim \mathcal{N}(0, \sigma^2 I),$$

$z \in \mathbb{R}^q$ and $\boldsymbol{W} \in \mathbb{R}^{d \times q}$. Then we have $x \sim \mathcal{N}(\boldsymbol{\mu}, \boldsymbol{W}\boldsymbol{W}^T + \sigma^2 I)$, $x|z \sim \mathcal{N}(\boldsymbol{W}z + \boldsymbol{\mu}, \sigma^2 I)$ and $z|x \sim \mathcal{N}(\boldsymbol{P}^{-1}\boldsymbol{W}^T(x - \boldsymbol{\mu}), \sigma^2 \boldsymbol{P}^{-1})$ where $\boldsymbol{P} = \boldsymbol{W}^T \boldsymbol{W} + \sigma^2 I$. The maximum likelihood estimator of $\boldsymbol{W}$ and $\sigma^2$ are

$$\boldsymbol{W}_{\mathrm{ML}} = \boldsymbol{U}_q (\boldsymbol{\Lambda}_q - \sigma_{\mathrm{ML}}^2 I)^{1/2} \quad \text{and} \quad \sigma_{\mathrm{ML}}^2 = \frac{1}{d - q} \sum_{i=q+1}^d \lambda_i,$$

where $\boldsymbol{U_q}$ is the matrix of the first $q$ columns of $\boldsymbol{U}$, $\boldsymbol{\Lambda_q}$ is the matrix of the first $q$ eigenvalues of $\boldsymbol{\Lambda}$. In the following study, we assume that $n$ is large enough such that we can learn the true $\boldsymbol{\mu}_*$ and $\boldsymbol{\Sigma}_*$. Thus we have $\boldsymbol{S} = \boldsymbol{\Sigma}_*$, $\boldsymbol{U}_q = \boldsymbol{U}_{q*}$, $\boldsymbol{\Lambda}_q = \boldsymbol{\Lambda}_{q*}$ for the generative model.

## 4.2 MINIMAX PROBLEM OF LATENT SPACE ADVERSARIAL TRAINING

To perturb the data in the latent space, Data will go through the encode-decode process $x \to z \to \Delta z + z \to x'$. Based on the probabilistic model, we may choose $z$ with the highest probability or just sample it from the distribution we learned. Hence, we could have different strategies. Below we list 3 strategies. Strategy 1 is used in practice and the other two are alternative choices. In lemma 1 we show that these strategies are equivalent under the low dimensional assumption. Hence, we do not need to worry about the effect of sampling strategies.

Strategy 1: Sample $x \sim \mathcal{D}$, encode $z = \arg\max q(z|x) = \boldsymbol{P}^{-1}\boldsymbol{W}^T(x - \boldsymbol{\mu}_*)$, add a perturbation $\Delta z$, and finally, decode $x_{adv} = \arg\max p(x|z + \Delta z) = \boldsymbol{W}(z + \Delta z) + \boldsymbol{\mu}_*$.

Strategy 2: Sample $x \sim \mathcal{D}$, then sample $z \sim q(z|x)$, add a perturbation $\Delta z$, and finally, sample $x_{adv} \sim p(x|z + \Delta z)$.

Strategy 3: Sample $z \sim \mathcal{N}(0, I)$, add a perturbation $\Delta z$, and then sample $x_{adv} \sim p(x|z + \Delta z)$. In this strategy, $x_{adv}$ can be viewed as the adversarial example of $x = \arg\max_x q(z|x)$.

The following lemma shows that the adversarial examples can be unified in one formula. Hence sampling strategies will not affect our analysis.

**Lemma 1** (Adversarial examples perturbed in the latent space). *Using these 3 strategies, the adversarial examples can be unified as*

$$x_{adv} = x' + \boldsymbol{W}\Delta z \quad and \quad x' \sim \mathcal{D}'_j = \mathcal{N}(\boldsymbol{\mu}_*, \boldsymbol{U}_* \boldsymbol{\Lambda}^{(j)} \boldsymbol{U}_*^T), \quad j = 1, 2, 3, \tag{5}$$

*where*

$$\boldsymbol{\Lambda}^{(1)} = \begin{bmatrix} (\boldsymbol{\Lambda_q} - \sigma^2 I)^2 \boldsymbol{\Lambda_q}^{-1} & 0 \\ 0 & 0 \end{bmatrix}, \quad \boldsymbol{\Lambda}^{(3)} = \begin{bmatrix} \boldsymbol{\Lambda_q} & 0 \\ 0 & \sigma^2 I \end{bmatrix},$$

$$\boldsymbol{\Lambda}^{(2)} = \begin{bmatrix} (\boldsymbol{\Lambda_q} - \sigma^2 I)^2 \boldsymbol{\Lambda_q}^{-1} + (\boldsymbol{\Lambda_q} - \sigma^2 I)\boldsymbol{\Lambda_q}^{-1}\sigma^2 + \sigma^2 I & 0 \\ 0 & \sigma^2 I \end{bmatrix}.$$

*If the data lie in a $q$ dimensional subspace, i.e. the covariance matrix $\boldsymbol{\Sigma}_*$ is rank $q$, we have $\boldsymbol{\Lambda}^{(1)} = \boldsymbol{\Lambda}^{(2)} = \boldsymbol{\Lambda}^{(3)} = \boldsymbol{\Lambda}_*$. Then $\mathcal{D}' = \mathcal{D}$.*

In general, the adversarial example can be decomposed into 2 parts, the change of distribution $x' \sim \mathcal{D}'$ and the small perturbation $\boldsymbol{W}\Delta z$. Therefore the adversarial expected risk can be written as the following minimax problem

$$\min_{\boldsymbol{\mu},\boldsymbol{\Sigma}} \mathcal{L}_{ls}(\boldsymbol{\mu}, \boldsymbol{\Sigma}; \mathcal{D}'_j) = \min_{\boldsymbol{\mu},\boldsymbol{\Sigma}} \mathbb{E}_{x' \sim \mathcal{D}'_j} \max_{\|\Delta z\| \le \varepsilon} \ell(x' + \boldsymbol{W}\Delta z, \boldsymbol{\mu}, \boldsymbol{\Sigma}), \quad j = 1, 2, 3. \tag{6}$$

We aim to analyze the different properties between the minimax problems in equations (4) and (6). We give the excess risk and optimal saddle point analysis in the following subsections.

## 4.3 EXCESS RISK ANALYSIS

We consider the difference between $\mathcal{L}_{ls}$ and $\mathcal{L}$ given the true $\boldsymbol{\Theta}_*$, i.e. $\mathcal{L}_{ls}(\boldsymbol{\Theta}_*; \mathcal{D}'_j) - \mathcal{L}(\boldsymbol{\Theta}_*; \mathcal{D})$. It characterizes the excess risk incured by the optimal perturbation. To derive the expression of excess risk, we decompose it into two parts

$$\mathcal{L}_{ls}(\boldsymbol{\Theta}_*; \mathcal{D}'_j) - \mathcal{L}(\boldsymbol{\Theta}_*; \mathcal{D}) = \underbrace{\mathcal{L}_{ls}(\boldsymbol{\Theta}_*; \mathcal{D}'_j) - \mathcal{L}(\boldsymbol{\Theta}_*; \mathcal{D}'_j)}_{perturbation} + \underbrace{\mathcal{L}(\boldsymbol{\Theta}_*; \mathcal{D}'_j) - \mathcal{L}(\boldsymbol{\Theta}_*; \mathcal{D})}_{change\ of\ distribution}. \tag{7}$$

To simplify the notation, we consider the Lagrange penalty form of the inner maximization problem in equation (6), i.e. $\max \ell(x' + \boldsymbol{W}\Delta z, \boldsymbol{\mu}, \boldsymbol{\Sigma}) - L\|\Delta z\|^2/2$, where $L$ is the Lagrange multiplier. The following theorem gives the solution in the general case.

**Theorem 2** (Excess risk). *Let $\mathcal{L}_{ls}$ and $\mathcal{L}$ be the loss with and without perturbation in latent space (equations (6) and (3) respectively), given the non-robustly learned $\boldsymbol{\Theta}_* = (\boldsymbol{\mu}_*, \boldsymbol{\Sigma}_*)$, thus the excess risk caused by perturbation is*

$$\mathcal{L}_{ls}(\boldsymbol{\Theta}_*, \mathcal{D}'_j) - \mathcal{L}(\boldsymbol{\Theta}_*, \mathcal{D}'_j) = \frac{1}{2}\sum_{i=1}^{q}\left[(1 + \frac{\lambda_i - \sigma^2}{(L-1)\lambda_i + \sigma^2})^2 - 1\right]\frac{\lambda_i^{(j)}}{\lambda_i}, \quad j = 1, 2, 3.$$

*and the excess risk caused by the changed of distribution is*

$$\mathcal{L}(\boldsymbol{\Theta}_*, \mathcal{D}'_j) - \mathcal{L}(\boldsymbol{\Theta}_*, \mathcal{D}) = \frac{1}{2}\log\left[\frac{\prod_{i=1}^{d}\lambda_i^{(j)}}{\prod_{i=1}^{d}\lambda_i}\right] + \frac{1}{2}\left(\sum_{i=1}^{d}\frac{\lambda_i^{(j)}}{\lambda_i} - d\right).$$

It is hard to see which part dominates the excess risk. If we further assume that the data lie in a $q$ dimensional manifold, the excess risk caused by the change of distribution becomes 0 by Lemma 1. We have the following corollary.

**Corollary 3** (Excess risk). *Let $\mathcal{L}_{ls}$ and $\mathcal{L}$ be the loss with or without perturbation in latent space (equation (6) and (3) respectively), given the non-robustly learned $\boldsymbol{\Theta}_* = (\boldsymbol{\mu}_*, \boldsymbol{\Sigma}_*)$, and $rank(\boldsymbol{\Sigma}_*) = q$. The excess risk*

$$\mathcal{L}_{ls}(\boldsymbol{\Theta}_*, \mathcal{D}'_j) - \mathcal{L}(\boldsymbol{\Theta}_*, \mathcal{D}) = \mathcal{O}(qL^{-2}).$$

The optimal perturbation in the latent space will incur an excess risk in $\mathcal{O}(qL^{-2})$. The adversarial vulnerability depends on the dimension $q$ and the Lagrange multiplier $L$. It does not depend on the shape of the data manifold. This is because the perturbation constraint (the black block) aligns with the shape of the data manifold (the ellipse) as we demonstrate in Figure 1 (c). Thus generative attacks will focus on the directions of the largest $q$ variance.

Then, we analyze the excess risk of original space adversarial training. Since the perturbation thresholds, $\varepsilon$, are on different scales in the original space attack and latent space attack, the corresponding Lagrange multipliers $L$ are different. We use $L'$ for original space adversarial training in the following Theorem.

**Theorem 4** (Excess risk of original space adversarial training). *Let $\mathcal{L}_r$ and $\mathcal{L}$ be the loss with or without perturbation in original space (equations (4) and (3) respectively), given the non-robustly learned $\boldsymbol{\Theta}_* = (\boldsymbol{\mu}_*, \boldsymbol{\Sigma}_*)$. Denote $\lambda_{min}$ be the smallest eigenvalue of $\boldsymbol{\Sigma}_*$. The excess risk is*

$$\Omega((\lambda_{min}L')^{-2}) \leq \mathcal{L}_r(\boldsymbol{\Theta}_*, \mathcal{D}) - \mathcal{L}(\boldsymbol{\Theta}_*, \mathcal{D}) \leq \mathcal{O}(d(\lambda_{min}L')^{-2}).$$

*If the data lie in a low dimensional manifold, i.e. $\lambda_{min} = 0$, the excess risk equals to $+\infty$.*

Optimal perturbation in the original space will incur an excess risk in $\mathcal{O}(d(\lambda_{min}L')^{-2})$. The adversarial vulnerability depends on the smallest eigenvalues $\lambda_{min}$, the dimension $d$, and the Lagrange multiplier $L'$. The $\lambda_{min}$ comes from the misalignment between the perturbation constraint (the black block) and the shape of the data manifold (the ellipse) as we demonstrate in Figure 1 (a). Notice that $\lambda_{min}$ also appears in the lower bound. Hence, the excess risk equals to $+\infty$ when $\lambda_{min} = 0$. Thus regular attacks focus on the directions of small variance. Specifically, when $\lambda_{min} = 0$, regular adversarial examples leave the data manifold.

### 4.4 SADDLE POINT ANALYSIS

In this subsection we study the optimal solution of optimization problem (6). Since it is not a standard minimax problem, we consider a modified problem:

$$\min_{\boldsymbol{\mu}, \boldsymbol{\Sigma}} \max_{\mathbb{E}_{x'} \|\Delta z\| = \varepsilon} \mathbb{E}_{x' \sim \mathcal{D}'_j} \ell(x' + \boldsymbol{W}\Delta z, \boldsymbol{\mu}, \boldsymbol{\Sigma}), \quad j = 1, 2, 3. \tag{8}$$

We will explain more about the connection between optimization problems in equation (6) and (8). See appendix A. The following theorem is our main result. It gives the optimal solution of latent space adversarial training.

**Theorem 5** (Main result: Optimal Saddle point). *The optimal solution of the modified problem in equation (8) is*

$$\boldsymbol{\mu}_{ls} = \boldsymbol{\mu}_* \quad and \quad \boldsymbol{\Sigma}_{ls} = \boldsymbol{U}_* \boldsymbol{\Lambda}^{ls} \boldsymbol{U}_*^T,$$

*where*

$$\lambda_i^{ls} = \frac{1}{4}\left[2\lambda_i^{(j)} + \frac{4(\lambda_i - \sigma^2)}{L} + 2\lambda_i^{(j)}\sqrt{1 + \frac{4(\lambda_i - \sigma^2)}{\lambda_i^{(j)}L}}\right] for\ i = 1 \leq q, \ \lambda_i^{ls} = \lambda_i^{(j)} for\ i > q,$$

*and $j = 1, 2, 3$ corresponding to strategies 1,2 and 3.*

We assume that the data lie in a $q$-dimensional manifold again. Then we have $\lambda_i^{ls}/\lambda_i = 1/2 + 1/L + \sqrt{1/4 + 1/L} \geq 1$ for $i \leq q$ and $\lambda_i^{ls}/\lambda_i = 0$ for $i > q$. Latent space adversarial training increases the model robustness by **amplifying large eigenvalues** of the data manifold. The illustration of the two dimensional case is in Figure 1, (c) and (d).

In the same setting, the optimal solution of standard adversarial training (Problem (4)) on the original space is in Theorem 6 (which is Theorem 2 in Ilyas et al. (2019)).

**Theorem 6** (Optimal saddle point Ilyas et al. (2019)). *The optimal solution of the problem in equation (4) is*

$$\boldsymbol{\mu}_r = \boldsymbol{\mu}_* \quad and \quad \boldsymbol{\Sigma}_r = \frac{1}{2}\boldsymbol{\Sigma}_* + \frac{1}{L'}I + \sqrt{\frac{1}{L'}\boldsymbol{\Sigma}_* + \frac{1}{4}\boldsymbol{\Sigma}_*^2}.$$

Theorem 6 is for the problem that the covariance matrix is restricted to be diagonal. Consider the ratio

$$\frac{\lambda_i^{(r)}}{\lambda_i} = \frac{1}{2} + \frac{1}{L'\lambda_i} + \sqrt{\frac{1}{L'\lambda_i} + \frac{1}{4}}.$$

For small true eigenvalue $\lambda_i$, the ratio is large. Standard adversarial training increases the robustness by **amplifying the small eigenvalues** of the data manifold. The illustration of the two dimensional case in Figure 1, (a) and (b).

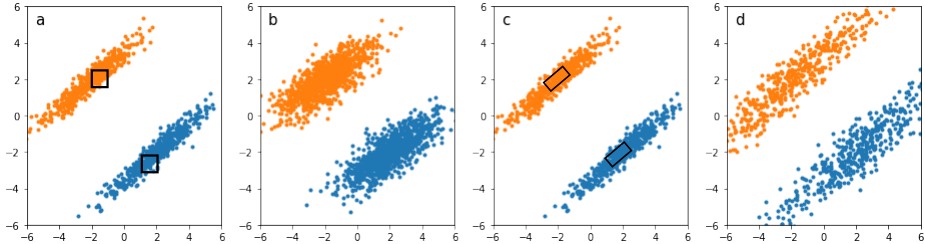

Figure 1: Demonstration of theoretical analysis: (a) Regular attacks directions; (b) Optimal saddle point of original space adversarial training; (c) Generative attacks directions; (d) Optimal saddle point of latent space adversarial training.

## 5 EXPERIMENTS

In this section we report our experimental results on training LeNet on MNIST (LeCun et al. (1998)) and ResNet (He et al. (2016)) on CIFAR-10 (Krizhevsky et al. (2009)) to confirm our theoretical findings. Both of the two datasets contain 50000 training samples and 10000 test samples. Details of the hyperparameters setting are in appendix B.

### 5.1 EIGENVALUES OF MNIST WITH AND WITHOUT ADVERSARIAL TRAINING

In this subsection, we show that the properties of the eigenvalues are also valid on the real dataset. We use MNIST as an example. Firstly, for each image in the test set $\mathcal{D}_{test}$ of MNIST, we vectorize it into a 784 dimension vector. Then we calculate the EVD (Eigenvalue decomposition) of the covariance matrix of each class. We plot the 784 eigenvalues of class 0 and 1 in Figure 2 in blue line as examples (the other 8 classes are shown in Appendix B.3). Secondly, we adversarially train a classifier $f_r(x)$ in original space. For each point $x$ in $\mathcal{D}_{test}$, we construct $x_r$ by PGD-attack with $\nabla_x f_r(x)$. Then we obtain a robust test set $\mathcal{D}_{test}^r$. We plot the intra class eigenvalues of the covariance matrix in Figure 2 in orange line. Lastly, we adversarially train a classifier $f_{ls}(x)$ in latent space. Then we construct latent space robust dataset $\mathcal{D}_{test}^{ls}$ by generative attack against $f_{ls}(x)$ on $\mathcal{D}_{test}$. The eigenvalues of the covariance matrix are shown in Figure 2 in green line.

We show all the eigenvalues in the first column in Figure 2. After adding a small perturbation in the original space or in the latent space, the distribution of $\mathcal{D}_{test}^r$ and $\mathcal{D}_{test}^{ls}$ are close to $\mathcal{D}_{test}$. We show the details of the figure of large eigenvalues (the first 30 eigenvalues) in the second column and small eigenvalues (last 754 eigenvalues) in the last column.

**Original space adversarial training focuses on the small variance direction** We adversarially train a robust classifier $f_r(x)$ in the original space. The robust test set $\mathcal{D}_{test}^r$ against $f_r(x)$ amplified the small eigenvalues a lot. In the third column, the orange line is significantly larger that the other 2 lines. While in the second column, the orange line is below the other two lines. The experiments give us an understanding of how the adversarial examples leave the data manifolds. The original dataset $\mathcal{D}_{test}$ lies in a low dimension affine plane in $\mathbb{R}^{784}$. After adversarial training, the data move towards the small variance directions. Finally $\mathcal{D}_{test}^r$ stay in a full dimension subspace.

**Latent space adversarial training focuses on the large variance direction** We adversarially train a robust classifier $f_{ls}(x)$ in the latent space. The latent space robust test set $\mathcal{D}_{test}^{ls}$ against $f_{ls}(x)$ amplified the large eigenvalues. In the second column, we can see that the green line is above the other two lines. And in the last column, the green line shows that adversarial training in the latent space will not affect the small eigenvalues. The adversarial examples in $\mathcal{D}_{test}^{ls}$ move along the large variance direction. Therefore, $\mathcal{D}_{test}^{ls}$ stay in the low dimension affine space.

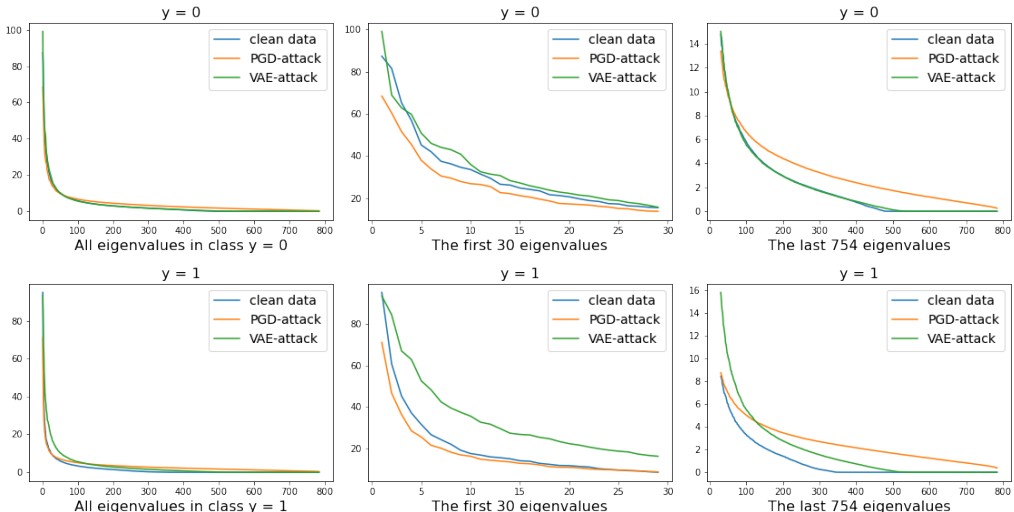

Figure 2: Eigenvalues of MNIST: The first row and the second row are of classes 0 and 1 respectively. The first column is about all the 784 eigenvalues of the dataset. The second column plots the large eigenvalues and the last column plots the small eigenvalues.

## 5.2 ROBUST TEST ACCURACIES

In this subsection, we compare the test accuracy of different attacks (FGSM, PGD, VAE) versus different defense (adversarial training with FGSM, PGD, and VAE) to help explain our theoretical results. We explain the results of MNIST in Table 1 as an example.

**On-manifold and off-manifold adversarial examples** The test accuracies of the standard training model on PGD and VAE-attack data are 3.9% and 42.4% respectively. Firstly, They show that on-manifold adversarial examples exist. Then, we can see that on-manifold adversarial examples are harder to find.

**Attack versus defense** Our theory tells us that original or latent space adversarial training increase the model robustness by amplifying the small or large eigenvalues, which are the variance of the distribution, in the data manifold. In experiments, the test accuracy of VAE-adv vs PGD-attack is 1.23%, which shows that extending the manifold boundary in directions of large variance gives no contribution to defending attacks in directions of small variance. Similarly, the test accuracy of PGD-adv vs VAE-attack is 52.18%, which shows that amplifying the boundary in the directions of small variance does not work well in defending attacks in directions of high variance. Further more, we can see that latent space adversarial training does not work well against a simple FGSM attack, and vice versa.

We see that PGD-adv can increase the test accuracies on VAE-attack from 42% to 52% on MNIST and from 19.40% to 26.31% on CIFAR-10. A possible reason is that original space adversarial training can amplify the small eigenvalues of the first $q$ dimension but fails to amplify the larger ones. As it is indicated in Figure 2, both original space and latent space adversarial training will increase the eigenvalues which are small but not equal to zero (around the 100th eigenvalue) of the covariance matrix.

## 5.3 ROBUSTNESS TRADE-OFF

Robustness trade-off are common in practice (Su et al. (2018); Tramèr and Boneh (2019)). For example, $\ell_1$ and $\ell_\infty$ adversarial examples cannot be defended simultaneously. The experiments on robustness trade-off are shown in Table 2. On MNIST, the jointly-trained model using both $\ell_1$ and $\ell_\infty$ adversarial examples will decrease the robust test accuracy by 20% comparing to the single trained model. On CIFAR-10, the jointly-trained model will decrease the test accuracies by 6% on $\ell_1$-attack and by 10% on $\ell_\infty$-attack. A possible reason is that they are all original space attacks and focus on

| MNIST | Std training | FGSM-Adv | PGD-Adv | VAE-Adv |
|---|---|---|---|---|
| clean data | **98.82%** | 98.79% | 98.73% | 98.28% |
| FGSM-Attack | 47.43% | 91.08% | **96.50%** | 27.94% |
| PGD-Attack | 3.90% | 15.46% | **95.51%** | 1.23% |
| VAE-Attack | 42.40% | 46.94% | 52.18% | **96.66%** |
| CIFAR-10 | Std training | FGSM-Adv | PGD-Adv | VAE-Adv |
| clean data | **92.14%** | 87.03% | 86.25% | 85.59% |
| FGSM-Attack | 38.39% | 76.92% | **77.34%** | 34.50% |
| PGD-Attack | 7.40% | 75.14% | **75.78%** | 10.53% |
| VAE-Attack | 19.40% | 31.80% | 26.31% | **40.18%** |

Table 1: Test accuracies of different attacks and defense algorithms on MNIST and CIFAR-10.

the same directions of small variance. Hence they conflict with each other. We study the robustness trade-off between regular and generative adversarial examples in this subsection.

| MNIST | $\ell_1$-Adv | $\ell_\infty$-Adv | joint | | | PGD-Adv | VAE-Adv | joint |
|---|---|---|---|---|---|---|---|---|
| $\ell_1$-attack | **78.50%** | 12.13% | 54.24% | | PGD-attack | **95.51%** | 1.23% | 89.50% |
| $\ell_\infty$-attack | 0.5% | **95.51%** | 78.81% | | VAE-attack | 52.18% | **96.66%** | 90.28% |
| CIFAR-10 | $\ell_1$-Adv | $\ell_\infty$-Adv | joint | | | PGD-Adv | VAE-Adv | joint |
| $\ell_1$-attack | **66.22%** | 16.41% | 60.23% | | PGD-attack | **75.78%** | 10.53% | 74.22% |
| $\ell_\infty$-attack | 52.8% | **75.78%** | 65.11% | | VAE-attack | 26.31% | 40.18% | **42.47%** |

Table 2: Comparison of the robustness trade-off between $\ell_1$ and $\ell_\infty$ attack and the trade-off between original space and latent space attack on MNIST and CIFAR-10.

Our theory suggests that adversarial robustness can be disentangled in different directions. Hence adversarial robustness against attacks on directions of low variance and large variance can be guaranteed simultaneously. On MNIST, the jointly-trained model decreases the test accuracies by 6% comparing to the single trained model. The conflict between on-manifold and off-manifold adversarial examples is much small than the conflict between off-manifold adversarial examples of different norms. On CIFAR-10, the jointly-trained model gets test accuracies 74% on PGD-attack and 42% on VAE-attack, which exhibit nearly no robustness trade-off. Therefore, if our goal is to defend mixture of regular and generative attacks, The jointly-train model will perform well.

**Discussion** Under the $q$-dimensional manifold assumption, there is an overlap between the directions of the $q$ largest variance and the directions of small variance. This is supported by Theorem 5, Theorem 6, and the first experiment. A possible reason for the robustness trade-off between regular and generative attacks is that they conflict with each other in the overlap directions. The conflict is not serious because they mainly focus on different directions.

## 6 CONCLUSION

In this paper, we show that adversarial robustness can be disentangled in directions of small variance and large variance of the data manifold. Theoretically, we study the excess risk and optimal saddle point of the minimax problem of latent space adversarial training. Experimentally, we show that these phenomena also exist in real datasets.

**Future works** One may design defense algorithms based on this property. We can generate adversarial examples based on the directions of the dataset without access to the model architecture. In white-box settings, we can use them for data augmentation to accelerate adversarial training or even increase the model robustness. However, we need to carefully compare the computational cost between calculating the EVD of the datasets and that of standard adversarial training. In black-box attacks, we can use them to query the target model. But the data manifolds should be transferred from some known datasets (since there is no information available for the training dataset of the target model). Theoretically, it is unclear whether we can find the closed-form solution under nonlinear models. How to analyze the nonlinear model is an open problem. Using linear models, we can further analyze the conflict between robustness and generalization.

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

## OVERVIEW

In appendix A, we provide the proof of the Theorems. In appendix B, we show the settings about the experiments. Appendix C is a further discussion of data augmentation using generative models. It is not closely related to our main paper.

## A   PROOF OF THE THEOREMS

### A.1   PROBLEM DESCRIPTION

**Lemma 1** (Adversarial examples perturbed in the latent space). *Using these 3 strategies, the adversarial examples can be unified as*

$$x_{adv} = x' + \boldsymbol{W}\Delta z \quad and \quad x' \sim \mathcal{D}'_j = \mathcal{N}(\boldsymbol{\mu}_*, \boldsymbol{U}_* \boldsymbol{\Lambda}^{(j)} \boldsymbol{U}_*^T), \quad j = 1, 2, 3,$$

*where*

$$\boldsymbol{\Lambda}^{(1)} = \begin{bmatrix} (\boldsymbol{\Lambda_q} - \sigma^2 I)^2 \boldsymbol{\Lambda_q}^{-1} & 0 \\ 0 & 0 \end{bmatrix} \quad \boldsymbol{\Lambda}^{(3)} = \begin{bmatrix} \boldsymbol{\Lambda_q} & 0 \\ 0 & \sigma^2 I \end{bmatrix}$$

$$\boldsymbol{\Lambda}^{(2)} = \begin{bmatrix} (\boldsymbol{\Lambda_q} - \sigma^2 I)^2 \boldsymbol{\Lambda_q}^{-1} + (\boldsymbol{\Lambda_q} - \sigma^2 I)\boldsymbol{\Lambda_q}^{-1}\sigma^2 + \sigma^2 I & 0 \\ 0 & \sigma^2 I \end{bmatrix}.$$

*If the data lie in a q dimensional subspace, i.e. the covariance matrix $\boldsymbol{\Sigma}_*$ is rank q, we have $\boldsymbol{\Lambda}^{(1)} = \boldsymbol{\Lambda}^{(2)} = \boldsymbol{\Lambda}^{(3)} = \boldsymbol{\Lambda}_*$. Then $\mathcal{D}' = \mathcal{D}$.*

Proof:

Recall that $x \sim \mathcal{N}(\boldsymbol{\mu}, \boldsymbol{W}\boldsymbol{W}^T + \sigma^2 I)$, $x|z \sim \mathcal{N}(\boldsymbol{W}z + \boldsymbol{\mu}, \sigma^2 I)$ and $z|x \sim \mathcal{N}(\boldsymbol{P}^{-1}\boldsymbol{W}^T(x - \boldsymbol{\mu}), \sigma^2 \boldsymbol{P}^{-1})$ where $\boldsymbol{P} = \boldsymbol{W}^T\boldsymbol{W} + \sigma^2 I$. The maximum likelihood estimator of $\boldsymbol{W}$ and $\sigma^2$ are

$$\boldsymbol{W}_{\mathrm{ML}} = \boldsymbol{U}_q(\boldsymbol{\Lambda}_q - \sigma^2 I)^{1/2} \quad and \quad \sigma^2_{\mathrm{ML}} = \frac{1}{d-q} \sum_{i=q+1}^{d} \lambda_i.$$

**Strategy 1**   Sample $x \sim \mathcal{D}$, encode $z = \arg\max q(z|x) = \boldsymbol{P}^{-1}\boldsymbol{W}^T(x - \boldsymbol{\mu}_*)$, add a perturbation $\Delta z$, and finally, decode $x_{adv} = \arg\max p(x|z + \Delta z) = \boldsymbol{W}(z + \Delta z) + \boldsymbol{\mu}_*$. Then

$$\begin{aligned} x_{adv} &= \boldsymbol{W}(\boldsymbol{P}^{-1}\boldsymbol{W}^T(x - \boldsymbol{\mu}_*) + \Delta z) + \boldsymbol{\mu}_* \\ &= \boldsymbol{W}\boldsymbol{P}^{-1}\boldsymbol{W}^T(x - \boldsymbol{\mu}_*) + \boldsymbol{\mu}_* + \boldsymbol{W}\Delta z \\ &= x' + \boldsymbol{W}\Delta z. \end{aligned}$$

Since $x \sim (\boldsymbol{\mu}_*, \boldsymbol{\Sigma}_*)$, we have $x - \boldsymbol{\mu}_* \sim (0, \boldsymbol{\Sigma}_*)$, Then

$$x' \sim \mathcal{N}(\boldsymbol{\mu}_*, \boldsymbol{W}\boldsymbol{P}^{-1}\boldsymbol{W}^T\boldsymbol{\Sigma}_*(\boldsymbol{W}\boldsymbol{P}^{-1}\boldsymbol{W}^T)^T),$$

With

$$\begin{aligned} &\boldsymbol{W}\boldsymbol{P}^{-1}\boldsymbol{W}^T\boldsymbol{\Sigma}_*(\boldsymbol{W}\boldsymbol{P}^{-1}\boldsymbol{W}^T)^T \\ =&\boldsymbol{U}_* \begin{bmatrix} \boldsymbol{\Lambda_q} - \sigma^2 & 0 \\ 0 & 0 \end{bmatrix}^{1/2} \begin{bmatrix} \boldsymbol{\Lambda_q} & 0 \\ 0 & \sigma^2 I \end{bmatrix}^{-1} \begin{bmatrix} \boldsymbol{\Lambda_q} - \sigma^2 & 0 \\ 0 & 0 \end{bmatrix}^{1/2} \boldsymbol{\Lambda}_* \\ &\begin{bmatrix} \boldsymbol{\Lambda_q} - \sigma^2 & 0 \\ 0 & 0 \end{bmatrix}^{1/2} \begin{bmatrix} \boldsymbol{\Lambda_q} & 0 \\ 0 & \sigma^2 I \end{bmatrix}^{-1} \begin{bmatrix} \boldsymbol{\Lambda_q} - \sigma^2 & 0 \\ 0 & 0 \end{bmatrix}^{1/2} \boldsymbol{U}_*^T \\ =&\boldsymbol{U}_* \begin{bmatrix} (\boldsymbol{\Lambda_q} - \sigma^2 I)^2 \boldsymbol{\Lambda_q}^{-1} & 0 \\ 0 & 0 \end{bmatrix} \boldsymbol{U}_*^T \\ =&\boldsymbol{U}_* \boldsymbol{\Lambda}^{(j)} \boldsymbol{U}_*^T, \quad j = 1. \end{aligned}$$

**Strategy 2** Sample $x \sim \mathcal{D}$, then sample $z \sim q(z|x)$, add a perturbation $\Delta z$, and finally, sample $x_{adv} \sim p(x|z + \Delta z)$. Then

$$z \sim \mathcal{N}(0, \boldsymbol{P}^{-1}\boldsymbol{W}^T\boldsymbol{\Sigma}_*(\boldsymbol{P}^{-1}\boldsymbol{W}^T)^T + \sigma^2\boldsymbol{P}^{-1})$$

and

$$x_{adv} \sim \mathcal{N}(\boldsymbol{\mu}_* + \boldsymbol{W}\Delta z, \boldsymbol{W}\boldsymbol{P}^{-1}\boldsymbol{W}^T\boldsymbol{\Sigma}_*(\boldsymbol{P}^{-1}\boldsymbol{W}^T)^T\boldsymbol{W}^T + \boldsymbol{W}\sigma^2\boldsymbol{P}^{-1}\boldsymbol{W}^T + \sigma^2 I),$$

$$x_{adv} = x' + \boldsymbol{W}\Delta z,$$

With

$$\boldsymbol{W}\boldsymbol{P}^{-1}\boldsymbol{W}^T\boldsymbol{\Sigma}_*(\boldsymbol{P}^{-1}\boldsymbol{W}^T)^T\boldsymbol{W}^T + \boldsymbol{W}\sigma^2\boldsymbol{P}^{-1}\boldsymbol{W}^T + \sigma^2 I$$

$$=\boldsymbol{U}_*\begin{bmatrix} (\boldsymbol{\Lambda}_q - \sigma^2 I)^2\boldsymbol{\Lambda}_q^{-1} & 0 \\ 0 & 0 \end{bmatrix}\boldsymbol{U}_*^T + \boldsymbol{U}_*\begin{bmatrix} (\boldsymbol{\Lambda}_q - \sigma^2 I)\boldsymbol{\Lambda}_q^{-1}\sigma^2 & 0 \\ 0 & 0 \end{bmatrix}\boldsymbol{U}_*^T + \sigma^2 I$$

$$=\boldsymbol{U}_*\begin{bmatrix} (\boldsymbol{\Lambda}_q - \sigma^2 I)^2\boldsymbol{\Lambda}_q^{-1} + (\boldsymbol{\Lambda}_q - \sigma^2 I)\boldsymbol{\Lambda}_q^{-1}\sigma^2 + \sigma^2 I & 0 \\ 0 & \sigma^2 I \end{bmatrix}\boldsymbol{U}_*^T$$

$$=\boldsymbol{U}_*\boldsymbol{\Lambda}^{(j)}\boldsymbol{U}_*^T, \quad j = 2.$$

**Strategy 3** Sample $z \sim \mathcal{N}(0, I)$, add a perturbation $\Delta z$, and then sample $x_{adv} \sim p(x|z + \Delta z)$. In this strategy, $x_{adv}$ can be viewed as the adversarial example of $x = \arg\max_x q(z|x)$.

$$x_{adv} \sim \mathcal{N}(\boldsymbol{\mu}_* + \boldsymbol{W}\Delta z, \boldsymbol{W}\boldsymbol{W}^T + \sigma^2 I),$$

With

$$\boldsymbol{W}\boldsymbol{W}^T + \sigma^2 I$$

$$=\boldsymbol{U}_*\begin{bmatrix} \boldsymbol{\Lambda}_q & 0 \\ 0 & \sigma^2 I \end{bmatrix}\boldsymbol{U}_*^T$$

$$=\boldsymbol{U}_*\boldsymbol{\Lambda}^{(j)}\boldsymbol{U}_*^T, \quad j = 3.$$

In these 3 strageties, the adversarial examples can be summerized as

$$x_{adv} = x' + \boldsymbol{W}\Delta z \quad \text{and} \quad x' \sim \mathcal{D}'_j, \quad j = 1, 2, 3,$$

where $j = 1, 2, 3$ corresponding to strategy 1,2 and 3.

If the data lie in a low dimensional space, i.e. the covariance matrix $\boldsymbol{\Sigma}_*$ is rank $q$. Then the maximum likelihood of $\sigma_{ML}^2 = \sum_{i=q+1}^d \lambda_i/(d - q) = 0$. Then

$$\boldsymbol{\Lambda}^{(1)} = \boldsymbol{\Lambda}^{(2)} = \boldsymbol{\Lambda}^{(3)} = \begin{bmatrix} \boldsymbol{\Lambda}_q & 0 \\ 0 & 0 \end{bmatrix} = \boldsymbol{\Lambda}_*.$$

There is no difference among these 3 strategies and there is no change of distribution, i.e. $\mathcal{D}' = \mathcal{D}$.

### A.1.1 EXCESS RISK ANALYSIS

Before we prove Theorem 2. We need to prove the following lemma first.

**Lemma 2** (optimal perturbation). *Given $\boldsymbol{\Theta} = (\boldsymbol{\mu}, \boldsymbol{\Sigma})$ the optimal solution of the inner max problem in equation 6 is*

$$\Delta z^* = \boldsymbol{W}^T(L\boldsymbol{\Sigma} - \boldsymbol{W}\boldsymbol{W}^T)^{-1}(x' - \boldsymbol{\mu}),$$

*where $L$ is the lagrange multiplier satisfying $\|\Delta z^*\| = \varepsilon$.*

Proof: Consider problem

$$\max_{\|\Delta z\| \leq \varepsilon} \ell(x' + \boldsymbol{W}\Delta z, \boldsymbol{\mu}, \boldsymbol{\Sigma}).$$

The Lagrangian function is

$$\ell(x' + \boldsymbol{W}\Delta z, \boldsymbol{\mu}, \boldsymbol{\Sigma}) - \frac{L}{2}(\|\Delta z\|^2 - \varepsilon^2)$$

$$=\frac{d}{2}\log(2\pi) + \frac{1}{2}\log|\boldsymbol{\Sigma}| + \frac{1}{2}(x' - \boldsymbol{\mu} + \boldsymbol{W}\Delta z)^T\boldsymbol{\Sigma}^{-1}(x' - \boldsymbol{\mu} + \boldsymbol{W}\Delta z) - \frac{L}{2}(\|\Delta z\|^2 - \varepsilon^2).$$

Notice that this quadratic objective function is concave when $L$ is larger than the largest eigenvalue of $\boldsymbol{W}^T \boldsymbol{\Sigma}^{-1} \boldsymbol{W}$. Calculate the partial derivative with respect to $\Delta z$ and set it to be zero, we have

$$\boldsymbol{W}^T \boldsymbol{\Sigma}^{-1} (x' - \boldsymbol{\mu} + \boldsymbol{W} \Delta z^*) - L \Delta z^* = 0$$
$$\Leftrightarrow (L - \boldsymbol{W}^T \boldsymbol{\Sigma}^{-1} \boldsymbol{W}) \Delta z^* = \boldsymbol{W}^T \boldsymbol{\Sigma}^{-1} (x' - \boldsymbol{\mu})$$
$$\Leftrightarrow \Delta z^* = (L - \boldsymbol{W}^T \boldsymbol{\Sigma}^{-1} \boldsymbol{W})^{-1} \boldsymbol{W}^T \boldsymbol{\Sigma}^{-1} (x' - \boldsymbol{\mu})$$
$$\Leftrightarrow \Delta z^* = \boldsymbol{W}^T (L \boldsymbol{\Sigma} - \boldsymbol{W} \boldsymbol{W}^T)^{-1} (x' - \boldsymbol{\mu}).$$

The last equation comes from the Woodbury matrix inversion Lemma. We can obtain $L$ by solving the equation $\|\Delta z^*\| = \varepsilon$. We don't have a closed form solution of $L$ but we can solve it numerically. $L \to \infty$ as $\varepsilon \to 0$. We only need to know $L$ is a constant in our whole theory.

**Theorem 2** (Excess risk). *Let $\mathcal{L}_{ls}$ and $\mathcal{L}$ be the loss with or without perturbation in latent space (equation 6 and 3 respectively), given the non-robustly learned $\boldsymbol{\Theta}_* = (\boldsymbol{\mu}_*, \boldsymbol{\Sigma}_*)$, The excess risk caused by perturbation is*

$$\mathcal{L}_{ls}(\boldsymbol{\Theta}_*, \mathcal{D}'_j) - \mathcal{L}(\boldsymbol{\Theta}_*, \mathcal{D}'_j) = \frac{1}{2} \sum_{i=1}^q \left[ (1 + \frac{\lambda_i - \sigma^2}{(L-1)\lambda_i + \sigma^2})^2 - 1 \right] \frac{\lambda_i^{(j)}}{\lambda_i}, \quad j = 1, 2, 3$$

*and the excess risk caused by changed of distribution is*

$$\mathcal{L}(\boldsymbol{\Theta}_*, \mathcal{D}'_j) - \mathcal{L}(\boldsymbol{\Theta}_*, \mathcal{D}) = \frac{1}{2} \log \left[ \frac{\prod_{i=1}^d \lambda_i^{(j)}}{\prod_{i=1}^d \lambda_i} \right] + \frac{1}{2} \left( \sum_{i=1}^d \frac{\lambda_i^{(j)}}{\lambda_i} - d \right).$$

Proof: Since
$$x' \sim \mathcal{D}_j = \mathcal{N}(\boldsymbol{\mu}_*, \boldsymbol{\Sigma_j}) = \mathcal{N}(\boldsymbol{\mu}_*, \boldsymbol{U}_* \boldsymbol{\Lambda}^{(j)} \boldsymbol{U}_*^T).$$

Denote
$$v = x' - \boldsymbol{\mu}_* \sim \mathcal{N}(0, \boldsymbol{U}_* \boldsymbol{\Lambda}^{(j)} \boldsymbol{U}_*^T).$$

And we have
$$\boldsymbol{W} \boldsymbol{W}^T = \boldsymbol{U_q}(\boldsymbol{\Lambda_q} - \sigma^2 I) \boldsymbol{U_q}^T = \boldsymbol{U}_* \begin{bmatrix} \boldsymbol{\Lambda_q} - \sigma^2 I & 0 \\ 0 & 0 \end{bmatrix} \boldsymbol{U}_*^T.$$

The excess risk caused by perturbation is

$$2(\mathcal{L}_{ls}(\boldsymbol{\Theta}_*, \mathcal{D}'_j) - \mathcal{L}(\boldsymbol{\Theta}_*, \mathcal{D}'_j))$$
$$= \mathbb{E}(v + \boldsymbol{W} \boldsymbol{W}^T (L\boldsymbol{\Sigma}_* - \boldsymbol{W} \boldsymbol{W}^T)^{-1} v)^T \boldsymbol{\Sigma}_*^{-1} (v + \boldsymbol{W} \boldsymbol{W}^T (L\boldsymbol{\Sigma}_* - \boldsymbol{W} \boldsymbol{W}^T)^{-1} v) - \mathbb{E} v^T \boldsymbol{\Sigma}_*^{-1} v$$
$$= Tr\left[ (I + \boldsymbol{W} \boldsymbol{W}^T (L\boldsymbol{\Sigma}_* - \boldsymbol{W} \boldsymbol{W}^T)^{-1})^T \boldsymbol{\Sigma}_*^{-1} (I + \boldsymbol{W} \boldsymbol{W}^T (L\boldsymbol{\Sigma}_* - \boldsymbol{W} \boldsymbol{W}^T)^{-1}) \mathbb{E} v v^T \right]$$
$$- Tr\left[ \boldsymbol{\Sigma}_*^{-1} \mathbb{E} v v^T \right]$$
$$= Tr\left[ \boldsymbol{U}_* \begin{bmatrix} [I + (\boldsymbol{\Lambda_q} - \sigma^2 I)((L-1)\boldsymbol{\Lambda_q} + \sigma^2 I)^{-1}]^2 & 0 \\ 0 & I \end{bmatrix} \boldsymbol{\Lambda}_*^{-1} \boldsymbol{\Lambda}^{(j)} \boldsymbol{U}_*^T \right] - Tr\left[ \boldsymbol{\Lambda}_*^{-1} \boldsymbol{\Lambda}^{(j)} \right]$$
$$= Tr\left[ \begin{bmatrix} [I + (\boldsymbol{\Lambda_q} - \sigma^2 I)((L-1)\boldsymbol{\Lambda_q} + \sigma^2 I)^{-1}]^2 & 0 \\ 0 & I \end{bmatrix} \boldsymbol{\Lambda}_*^{-1} \boldsymbol{\Lambda}^{(j)} \right] - Tr\left[ \boldsymbol{\Lambda}_*^{-1} \boldsymbol{\Lambda}^{(j)} \right]$$
$$= \sum_{i=1}^q \left[ (1 + \frac{\lambda_i - \sigma^2}{(L-1)\lambda_i + \sigma^2})^2 - 1 \right] \frac{\lambda_i^{(j)}}{\lambda_i}, \quad j = 1, 2, 3.$$

and the excess risk caused by changed of distribution is

$$2(\mathcal{L}(\boldsymbol{\Theta}_*, \mathcal{D}'_j) - \mathcal{L}(\boldsymbol{\Theta}_*, \mathcal{D}))$$
$$= \log |\boldsymbol{\Sigma_j}| - \log |\boldsymbol{\Sigma}_*| + \mathbb{E}_{x'}(x' - \boldsymbol{\mu}_*)^T \boldsymbol{\Sigma}_*^{-1}(x' - \boldsymbol{\mu}_*) - \mathbb{E}_x(x - \boldsymbol{\mu}_*)^T \boldsymbol{\Sigma}_*^{-1}(x - \boldsymbol{\mu}_*)$$
$$= \log |\boldsymbol{\Sigma_j}| - \log |\boldsymbol{\Sigma}_*| + Tr(\boldsymbol{\Sigma}_*^{-1} \mathbb{E}_{x'}(x' - \boldsymbol{\mu}_*)(x' - \boldsymbol{\mu}_*)^T) - Tr(\boldsymbol{\Sigma}_*^{-1} \mathbb{E}_x(x - \boldsymbol{\mu}_*)(x - \boldsymbol{\mu}_*)^T)$$
$$= \log \left[ \frac{\prod_{i=1}^d \lambda_i^{(j)}}{\prod_{i=1}^d \lambda_i} \right] + Tr(\boldsymbol{\Lambda}_*^{-1} \boldsymbol{\Lambda}^{(j)}) - Tr(\boldsymbol{\Lambda}_*^{-1} \boldsymbol{\Lambda}_*)$$
$$= \log \left[ \frac{\prod_{i=1}^d \lambda_i^{(j)}}{\prod_{i=1}^d \lambda_i} \right] + \left( \sum_{i=1}^d \frac{\lambda_i^{(j)}}{\lambda_i} - d \right).$$

$\square$

It is hard to see which part dominates the excess risk. If we further assume that the data lie in a $q$ dimension manifold. The excess risk caused by the change of distribution becomes $0$. We have the following corollary.

**Corollary 3** (Excess risk). *Let $\mathcal{L}_{ls}$ and $\mathcal{L}$ be the loss with or without perturbation in latent space (equation (6) and (3) respectively), given the non-robustly learned $\mathbf{\Theta}_* = (\boldsymbol{\mu}_*, \boldsymbol{\Sigma}_*)$, and $rank(\boldsymbol{\Sigma}_*) = q$. The excess risk*

$$\mathcal{L}_{ls}(\mathbf{\Theta}_*, \mathcal{D}'_j) - \mathcal{L}(\mathbf{\Theta}_*, \mathcal{D}) = \mathcal{O}(qL^{-2}).$$

Proof: By Lemma 1, we have $\sigma^2 = 0$. $\lambda_i^{(j)} = \lambda_i$ and $\mathcal{D}'_j = \mathcal{D}$. Hence the excess risk caused by changed of distribution

$$\mathcal{L}(\mathbf{\Theta}_*, \mathcal{D}'_j) - \mathcal{L}(\mathbf{\Theta}_*, \mathcal{D}) = 0.$$

The excess risk caused by perturbation is

$$
\begin{aligned}
& 2(\mathcal{L}_{ls}(\mathbf{\Theta}_*, \mathcal{D}'_j) - \mathcal{L}(\mathbf{\Theta}_*, \mathcal{D}'_j)) \\
= & \sum_{i=1}^{q} \Big[(1 + \frac{\lambda_i - \sigma^2}{(L-1)\lambda_i + \sigma^2})^2 - 1\Big]\frac{\lambda_i^{(j)}}{\lambda_i} \\
= & \sum_{i=1}^{q} \Big[(1 + \frac{1}{(L-1)})^2 - 1\Big] \\
= & \mathcal{O}(qL^{-2}).
\end{aligned}
$$

$\square$

**Theorem 4** (Excess risk of orginal space adversarial training). *Let $\mathcal{L}_r$ and $\mathcal{L}$ be the loss with or without perturbation in original space (equations (4) and (3) respectively), given the non-robustly learned $\mathbf{\Theta}_* = (\boldsymbol{\mu}_*, \boldsymbol{\Sigma}_*)$. Denote $\lambda_{min}$ be the smallest eigenvalue of $\boldsymbol{\Sigma}_*$. The excess risk*

$$\Omega((\lambda_{min}L)^{-2}) \leq \mathcal{L}_r(\mathbf{\Theta}_*, \mathcal{D}) - \mathcal{L}(\mathbf{\Theta}_*, \mathcal{D}) \leq \mathcal{O}(d(\lambda_{min}L)^{-2}).$$

Theorem 4 can be viewed as a corollary of Theorem 1 in Ilyas et al. (2019). We give the prove here.

Proof: Consider the Lagrange multiplier form of the inner maximization problem in equation 4.

$$\max_{\|\Delta x\| \leq \varepsilon} \ell(x + \Delta x, \boldsymbol{\mu}, \boldsymbol{\Sigma}).$$

The Lagrangian function is

$$
\begin{aligned}
& \ell(x + \Delta x, \boldsymbol{\mu}, \boldsymbol{\Sigma}) - \frac{L}{2}(\|\Delta x\|^2 - \varepsilon^2) \\
= & \frac{d}{2}\log(2\pi) + \frac{1}{2}\log|\boldsymbol{\Sigma}| + \frac{1}{2}(x - \boldsymbol{\mu} + \Delta x)^T\boldsymbol{\Sigma}^{-1}(x - \boldsymbol{\mu} + \Delta x) - \frac{L}{2}(\|\Delta x\|^2 - \varepsilon^2).
\end{aligned}
$$

Notice that this quadratic objective function is concave when $L$ is larger than the largest eigenvalue of $\boldsymbol{\Sigma}^{-1}$. Calculate the partial derivative with respect to $\Delta x$ and set it to be zero, we have

$$
\begin{aligned}
& \boldsymbol{\Sigma}^{-1}(x - \boldsymbol{\mu} + \Delta x^*) - L\Delta x^* = 0 \\
\Leftrightarrow & \Delta x^* = (L\boldsymbol{\Sigma} - I)^{-1}(x - \boldsymbol{\mu}).
\end{aligned}
$$

The excess risk is

$$
\begin{aligned}
& 2(\mathcal{L}_r(\mathbf{\Theta}_*, \mathcal{D}) - \mathcal{L}(\mathbf{\Theta}_*, \mathcal{D})) \\
= & \mathbb{E}(v + (L\boldsymbol{\Sigma}_* - I)^{-1}v)^T\boldsymbol{\Sigma}_*^{-1}(v + (L\boldsymbol{\Sigma}_* - I)^{-1}v) - \mathbb{E}v^T\boldsymbol{\Sigma}_*^{-1}v \\
= & Tr\big[(I + (L\boldsymbol{\Sigma}_* - I)^{-1})^T\boldsymbol{\Sigma}_*^{-1}(I + (L\boldsymbol{\Sigma}_* - I)^{-1})\mathbb{E}vv^T\big] - Tr\big[\boldsymbol{\Sigma}_*^{-1}\mathbb{E}vv^T\big] \\
= & \sum_{i=1}^{d}[(1 + \frac{1}{L\lambda_i - 1})^2 - 1].
\end{aligned}
$$

On the one hand,

$$\sum_{i=1}^{d}[(1 + \frac{1}{L\lambda_i - 1})^2 - 1]$$

$$\geq[(1 + \frac{1}{L\lambda_{min} - 1})^2 - 1]$$

$$\geq\Omega((L\lambda_{min})^{-2}).$$

On the other hand,

$$\sum_{i=1}^{d}[(1 + \frac{1}{L\lambda_i - 1})^2 - 1]$$

$$\leq d[(1 + \frac{1}{L\lambda_{min} - 1})^2 - 1]$$

$$\leq\mathcal{O}(d(L\lambda_{min})^{-2}).$$

$\square$

### A.1.2 SADDLE POINT ANALYSIS

**Theorem 5** (Main result: Optimal Saddle point). *The optimal solution of the modified problem in equation (8) is*

$$\boldsymbol{\mu}_{ls} = \boldsymbol{\mu}_* \quad and \quad \boldsymbol{\Sigma}_{ls} = \boldsymbol{U}_*\boldsymbol{\Lambda}^{ls}\boldsymbol{U}_*^T,$$

*where*

$$\lambda_i^{ls} = \frac{1}{4}\Big[2\lambda_i^{(j)} + \frac{4(\lambda_i - \sigma^2)}{L} + 2\lambda_i^{(j)}\sqrt{1 + \frac{4(\lambda_i - \sigma^2)}{\lambda_i^{(j)}L}}\Big] \text{ for } i = 1 \leq q \text{ and } \lambda_i^{ls} = \lambda_i^{(j)} \text{ for } i > q.$$

$j = 1, 2, 3$ *corresponding to strategies 1,2 and 3.*

Problem 6 is not a standard minimax problem, consider the modified problem

$$\min_{\boldsymbol{\mu},\boldsymbol{\Sigma}} \max_{\mathbb{E}_{x'}\|\Delta z\|=\varepsilon} \mathbb{E}_{x'\sim\mathcal{D}'_j}\ell(x' + \boldsymbol{W}\Delta z, \boldsymbol{\mu}, \boldsymbol{\Sigma}), \quad j = 1, 2, 3. \tag{9}$$

By lemma 3, the optimal perturbation $\Delta z^*$ is a matrix $M$ times $x - \boldsymbol{\mu}$. Consider the problem

$$\min_{\boldsymbol{\mu},\boldsymbol{\Sigma}} \max_{\mathbb{E}_{x'}\|M(x'-\boldsymbol{\mu})\|=\varepsilon} \mathbb{E}_{x'\sim\mathcal{D}'_j}\ell(x' + \boldsymbol{W}M(x' - \boldsymbol{\mu}), \boldsymbol{\mu}, \boldsymbol{\Sigma}), \quad j = 1, 2, 3. \tag{10}$$

**Lemma 6** (optimal perturbation). *Given $\boldsymbol{\Theta} = (\boldsymbol{\mu}, \boldsymbol{\Sigma})$ the optimal solution of the inner max problem of 10 is*

$$M^* = \boldsymbol{W}^T(L\boldsymbol{\Sigma} - \boldsymbol{W}\boldsymbol{W}^T)^{-1}.$$

Proof: Consider the problem

$$\max_{\mathbb{E}\|M(x'-\boldsymbol{\mu})\|=\varepsilon} \mathbb{E}\ell(x' + \boldsymbol{W}M(x' - \boldsymbol{\mu}), \boldsymbol{\mu}, \boldsymbol{\Sigma})).$$

The lagrangian function is

$$\mathbb{E}\Big[\ell(x' + \boldsymbol{W}M(x' - \boldsymbol{\mu}), \boldsymbol{\mu}, \boldsymbol{\Sigma}) - \frac{L}{2}(\|M(x' - \boldsymbol{\mu})\|^2 - \varepsilon^2)\Big].$$

Let $x' - \boldsymbol{\mu} = v$, Take the gradient with respect to $M$ and set it to be zero, we have

$$\frac{\partial}{\partial M}\mathbb{E}\Big[\ell(x' + \boldsymbol{W}M(x' - \boldsymbol{\mu}), \boldsymbol{\mu}, \boldsymbol{\Sigma}) - \frac{L}{2}(\|M(x' - \boldsymbol{\mu})\|^2 - \varepsilon^2)\Big]$$

$$=\nabla_M\mathbb{E}\Big[v^T M\boldsymbol{W}^T\boldsymbol{\Sigma}^{-1}v + \frac{1}{2}v^T M\boldsymbol{W}^T\boldsymbol{\Sigma}^{-1}\boldsymbol{W}Mv - Lv^T MMv/2\Big]$$

$$=\Big[\boldsymbol{W}^T\boldsymbol{\Sigma}^{-1} + \boldsymbol{W}^T\boldsymbol{\Sigma}^{-1}\boldsymbol{W}M - LM\Big]\mathbb{E}[vv^T]$$

$$=0.$$

Then we have

$$M^* = (L - \boldsymbol{W}^T \boldsymbol{\Sigma}^{-1} \boldsymbol{W})^{-1} \boldsymbol{W}^T \boldsymbol{\Sigma}^{-1}$$
$$= \boldsymbol{W}^T (L \boldsymbol{\Sigma} - \boldsymbol{W} \boldsymbol{W}^T)^{-1}.$$

$\square$

The last equality is the Woodbury matrix inversion Lemma. Notice that lemma 3 and Lemma 6 have the same form of solution. This is why we can use Problem 10 to approximate Problem 6. To solve the problem 10, we need to introduce Danskin's Theorem.

**Theorem 7** (Danskin's Theorem). *Suppose $\phi(x, z) : \mathcal{X} \times \mathcal{Z} \to \mathbb{R}$ is a continuous function of two arguments, where $\mathcal{Z} \subset \mathbb{R}^m$ is compact. Define $f(x) = \max_{z \in \mathcal{Z}} \phi(x, z)$. Then, if for every $z \in \mathcal{Z}$, $\phi(x, z)$ is convex and differentiable in $x$, and $\partial \phi / \partial x$ is continuous:*
*The subdifferential of $f(x)$ is given by*

$$\partial f(x) = conv\{\frac{\partial \phi(x, z)}{\partial x}, z \in \mathcal{Z}_0(x)\},$$

*where $conv(\cdot)$ is the convex hull, and $\mathcal{Z}_0(x)$ is*

$$\mathcal{Z}_0(x) = \{\bar{z} : \phi(x, \bar{z}) = \max \phi(x, z)\}.$$

If the outer minimization problem is convex and differentiable, we can use any maximizer for the inner maximization problem to find the saddle point. But the outer problem of problem 4 is not convex, we need to modify the problem again. Assume that we have already obtained the eigenvector $\boldsymbol{U}^*$ from the ML estimator. The optimization variables of the outer minimization problem are $\boldsymbol{\mu}$ and $\boldsymbol{\Lambda}$. Then we have $\boldsymbol{\Sigma} = \boldsymbol{U}_* \boldsymbol{\Lambda} \boldsymbol{U}_*^T$ in problem 10. Another reason to make this assumption is that we only need to consider the eigenvalue problem to compare with standard adversarial training (Theorem 2 of Andrew Ilyas et al., 2019).

Proof of Theorem 5:

By Lemma 6, we have

$$M^* = \boldsymbol{W}^T (L\boldsymbol{\Sigma} - \boldsymbol{W}\boldsymbol{W}^T)^{-1}$$
$$= \begin{bmatrix} \boldsymbol{\Lambda}_q - \sigma^2 & 0 \\ 0 & 0 \end{bmatrix}^{1/2} \left( L\boldsymbol{\Lambda} - \begin{bmatrix} \boldsymbol{\Lambda}_q - \sigma^2 & 0 \\ 0 & 0 \end{bmatrix} \right)^{-1} \boldsymbol{U}_*^T.$$

Which is a diagonal matrix $\boldsymbol{\Lambda}_M$ times $\boldsymbol{U}_*^T$. Let $T = \boldsymbol{\Lambda}^{-1}$, $m = \boldsymbol{\Lambda}^{-1} \boldsymbol{U}_*^T \boldsymbol{\mu}$ and $x'' = \boldsymbol{U}_*^T x'$. The optimization problem becomes

$$\min_{m,T} \max_{\boldsymbol{\Lambda}_M} \quad \mathbb{E}_{x' \sim \mathcal{D}_j'} \ell(x' + \boldsymbol{W}\boldsymbol{\Lambda}_M \boldsymbol{U}_*^T(x' - \boldsymbol{\mu}), m, T) \tag{11}$$
$$\text{s.t.} \quad \mathbb{E}_{x'} \|\boldsymbol{\Lambda}_M \boldsymbol{U}_*(x' - \boldsymbol{\mu})\|^2 = \varepsilon^2.$$

Obviously, the inner constraint is compact (by Heine-Borel theorem), we only need to prove the convexity of the outer problem to use Danskin's Theorem. For any $x'$ and $\boldsymbol{\Lambda}_M$,

$$\ell(x' + \boldsymbol{W}\boldsymbol{\Lambda}_M^T \boldsymbol{U}_*(x' - \boldsymbol{\mu}), m, T)$$
$$= \frac{d}{2}\log(2\pi) + \frac{1}{2}\log|\boldsymbol{\Sigma}| + \frac{1}{2}(x' - \boldsymbol{\mu} + \boldsymbol{W}M(x' - \boldsymbol{\mu})^T \boldsymbol{\Sigma}^{-1}(x' - \boldsymbol{\mu} + \boldsymbol{W}M(x' - \boldsymbol{\mu}).$$

Let $u = \boldsymbol{U}_*^T(x' - \boldsymbol{\mu})$, and $A = (I + \begin{bmatrix} \boldsymbol{\Lambda}_q - \sigma^2 & 0 \\ 0 & 0 \end{bmatrix} \boldsymbol{\Lambda}_M)^2$, consider the third term, we have

$$\frac{1}{2}\log|\boldsymbol{\Sigma}| + \frac{1}{2}(x' - \boldsymbol{\mu} + \boldsymbol{W}M(x' - \boldsymbol{\mu})^T \boldsymbol{\Sigma}^{-1}(x' - \boldsymbol{\mu} + \boldsymbol{W}M(x' - \boldsymbol{\mu})$$
$$= \frac{1}{2}u^T A^2 T u.$$

By Daskalakis et al., 2018Daskalakis et al. (2018), The hessian matrix is

$$H = \text{Cov}_{z \sim \mathcal{N}(T^{-1}m, (AT)^{-1})}\left[ \begin{pmatrix} \text{vec}(-\frac{1}{2}Azz^T) \\ z \end{pmatrix}, \begin{pmatrix} \text{vec}(-\frac{1}{2}Azz^T) \\ z \end{pmatrix} \right] \succeq 0.$$

Therefore, this is a convex problem. By the same calculation in Lemma 6, a maximizer of the inner problem is

$$\mathbf{\Lambda}_M^* = \begin{bmatrix} \mathbf{\Lambda_q} - \sigma^2 & 0 \\ 0 & 0 \end{bmatrix}^{1/2} \left( L\mathbf{\Lambda} - \begin{bmatrix} \mathbf{\Lambda_q} - \sigma^2 & 0 \\ 0 & 0 \end{bmatrix} \right)^{-1} \quad.$$

Then

$$A = \left[ I + \begin{bmatrix} \mathbf{\Lambda_q} - \sigma^2 & 0 \\ 0 & 0 \end{bmatrix} \left( L\mathbf{\Lambda} - \begin{bmatrix} \mathbf{\Lambda_q} - \sigma^2 & 0 \\ 0 & 0 \end{bmatrix} \right)^{-1} \right]^2.$$

Then the first order derivative (by Daskalakis et al. (2018)) is

$$\nabla_{[T,m]^T} \ell = \begin{bmatrix} \frac{1}{2} A\mathbf{\Lambda}^{(j)} - \frac{1}{2}T^{-1} \\ AT^{-1}m - A\mathbf{U}_*^T \boldsymbol{\mu}_* \end{bmatrix} = 0.$$

From the second equation, we directly have $\boldsymbol{\mu}_{ls} = \boldsymbol{\mu}_*$. From the first equation, for $i > q$, we have

$$(1+0)^2 \lambda_i^{(j)} = \lambda_i^{ls}.$$

For $i \le q$, we have

$$(1 + (\lambda_i - \sigma^2)/(L\lambda_i^{ls} - \lambda_i + \sigma^2))^2 \lambda_i^{(j)} = \lambda_i^{ls}.$$

It equivalents to a second order equation of $\sqrt{\lambda_i^{ls}}$

$$\sqrt{\lambda_i^{ls}}^2 - \sqrt{\lambda_i^{(j)}}\sqrt{\lambda_i^{ls}} - \frac{\lambda_i - \sigma^2}{L} = 0.$$

Solving this equation, we obtained

$$\lambda_i^{ls} = \frac{1}{4}\left[ 2\lambda_i^{(j)} + \frac{4(\lambda_i - \sigma^2)}{L} + 2\lambda_i^{(j)}\sqrt{1 + \frac{4(\lambda_i - \sigma^2)}{\lambda_i^{(j)}L}} \right] \text{ for } i = 1 \le q \text{ and } \lambda_i^{ls} = \lambda_i^{(j)}, \text{ for } i > q.$$

$\square$

# B  EXPERIMENTS SETTINGS

## B.1  MNIST

For Mnist, we use LeNet5 for the classifier and 2 layers MLP (with hidden size 256 and 784) for the encoder and decoder of conditional VAE. For standard training of the classifier, we use 30 epochs, batch size 128, learning rate $10^{-3}$, and weight decay $5 \times 10^{-4}$. For the CVAE, we use 20 epochs, learning rate $10^{-3}$, batch size 64, and latent size 10.

For standard adversarial training, we use $\varepsilon = 0.25$ for FGSM and PGD. in PGD, we use 40 steps for the inner part. Adversarial training start after 10 epochs standard training.

For generative adversarial training, we use $\varepsilon = 1$ in the latent space with FGSM. Adversarial training start after 10 epoches standard training.

In the attack part, we use $\varepsilon = 0.2$ for norm-based attack and $\varepsilon = 1$ for generative attack on the test set.

## B.2  CIFAR10

For CIFAR10, we use ResNet32 for the classifier and 4 layers CNN for the encoder and decoder of conditional VAE. For standard training of the classifier, we use 200 epochs, batch size 128, learning rate $10^{-3}$, and weight decay $5 \times 10^{-4}$. For the CVAE, we use 100 epochs, learning rate $10^{-3}$, batch size 64, and latent size 128.

For standard adversarial training, we use $\varepsilon = 4/255$ for FGSM and PGD. in PGD, we use 10 steps for the inner part. Adversarial training start after 100 epochs standard training.

For generative adversarial training, we use $\varepsilon = 0.1$ in the latent space with FGSM. Adversarial training start after 100 epoches standard training. Since we see that the modeling power of VAE in CIFAR10 is not good enough. For each of the image, the encode variance is very small. When we

add a small perturbation to the encode mean value, the output image are blured. Hence we only use a small $\varepsilon = 0.1$.

In the attack part, we use $\varepsilon = 4/255$ for norm-based attacks and $\varepsilon = 0.1$ for generative attack on the test set. The test accuracy of VAE-adv against VAE attack is $40.18\%$ in our experiments. It is not good enough because of the modeling power of VAE. But the results on standard adversarial training versus VAE-attacks are worse, and vice versa. The experiments support our findings.

### B.3 EIGENVALUES OF COVARIANCE MATRIX OF MNIST

We plot the eigenvalues of all the classes in this section, see Figure 3.

### C FINITE SAMPLES CASE: DATA AUGMENTATION OR ADVERSARIAL TRAINING

In this section we discuss the question that can we use the generative model to generate more examples for training in our theoretical framework. Because it is not closely related to our main results, we only discuss it in appendix. Let us focus on the case that the number of samples are not enough.

**Generative model cannot help data augmentation** Given dataset $\{x_i\}_{i=1}^n \subset \mathbb{R}^d$, let $\hat{\boldsymbol{\mu}}$ and $\hat{\boldsymbol{S}} = \hat{\boldsymbol{U}}\hat{\boldsymbol{\Lambda}}\hat{\boldsymbol{U}}$ be the sample mean and sample covariance matrix. The generative model learned by this dataset is $x = \hat{\boldsymbol{U}}_q(\hat{\boldsymbol{\Lambda}}_q - \sigma^2 I)^{1/2}z + \hat{\boldsymbol{\mu}} + \boldsymbol{\epsilon}$. The distribution of the data sample from the model is

$$x_{gen} \sim \mathcal{N}(\hat{\boldsymbol{\mu}}, \boldsymbol{W}\boldsymbol{W}^T + \sigma^2 I) = \mathcal{N}(\hat{\boldsymbol{\mu}}, \hat{\boldsymbol{U}}\hat{\boldsymbol{\Lambda}}_{gen}\hat{\boldsymbol{U}}) \triangleq \mathcal{N}(\hat{\boldsymbol{\mu}}, \hat{\boldsymbol{S}}'). \tag{12}$$

If we use $n$ samples from the original dataset and $m$ samples from the generative model, the loss function is

$$\mathcal{L}(\boldsymbol{\mu}, \boldsymbol{\Sigma}) = \min_{\boldsymbol{\mu}, \boldsymbol{\Sigma}} \frac{1}{n+m}\sum_{i=1}^n \ell(x_i; \boldsymbol{\mu}, \boldsymbol{\Sigma}) + \frac{m}{n+m}\mathbb{E}_{x \sim \mathcal{N}(\hat{\boldsymbol{\mu}}, \hat{\boldsymbol{S}}')}[\ell(x; \boldsymbol{\mu}, \boldsymbol{\Sigma})]. \tag{13}$$

**Theorem 8** (Data augmentation by generative model). *Given MLE $\hat{\boldsymbol{\mu}}$ and $\hat{\boldsymbol{S}}$, The optimal solution training with $n$ true samples and $m$ generated samples is $\boldsymbol{\mu}_{da} = \hat{\boldsymbol{\mu}}$ and $\boldsymbol{\Sigma}_{da} = \hat{\boldsymbol{U}}\boldsymbol{\Lambda}^{da}\hat{\boldsymbol{U}}^T$, where*

$$\lambda_i^{da} = \hat{\lambda}_i \quad for \quad i \le q \quad and \quad \lambda_i^{da} = \frac{n}{n+m}\hat{\lambda}_i + \frac{m}{(n+m)(d-q)}\sum_{k=q+1}^d \hat{\lambda}_k \quad for \quad i > q.$$

Proof: Consider the derivative of Problem 13.

$$\nabla_{\boldsymbol{\mu}}\mathcal{L} = \frac{1}{n+m}\sum_{i=1}^n \boldsymbol{\Sigma}^{-1}(x_i - \boldsymbol{\mu}) + \frac{m}{n+m}\mathbb{E}_x \boldsymbol{\Sigma}^{-1}(x - \boldsymbol{\mu}) = 0.$$

Then we can obtain $\boldsymbol{\mu}_{da} = \hat{\boldsymbol{\mu}}$.

$$\nabla_{\boldsymbol{\Sigma}^{-1}}\mathcal{L} = -\boldsymbol{\Sigma} + \frac{1}{n+m}\sum_{i=1}^n (x_i - \boldsymbol{\mu})(x_i - \boldsymbol{\mu})^T + \frac{m}{n+m}\mathbb{E}_x(x - \boldsymbol{\mu})(x - \boldsymbol{\mu})^T = 0.$$

Then we have

$$\boldsymbol{\Sigma}_{ls} = \frac{n}{n+m}\hat{\boldsymbol{S}} + \frac{m}{n+m}\hat{\boldsymbol{S}}'.$$

For $i \le q$,

$$\lambda_i^{ls} = \frac{n}{n+m}\hat{\lambda}_i + \frac{m}{n+m}\hat{\lambda}_i = \hat{\lambda}_i.$$

For $i > q$,

$$\lambda_i^{ls} = \frac{n}{n+m}\hat{\lambda}_i + \frac{m}{n+m}\hat{\sigma^2} = \frac{n}{n+m}\hat{\lambda}_i + \frac{m}{(n+m)(d-q)}\sum_{k=q+1}^d \hat{\lambda}_k.$$

$\square$

The optimal solution is a little bit destoryed. In this perspective, generative models give no help to data augumentation.

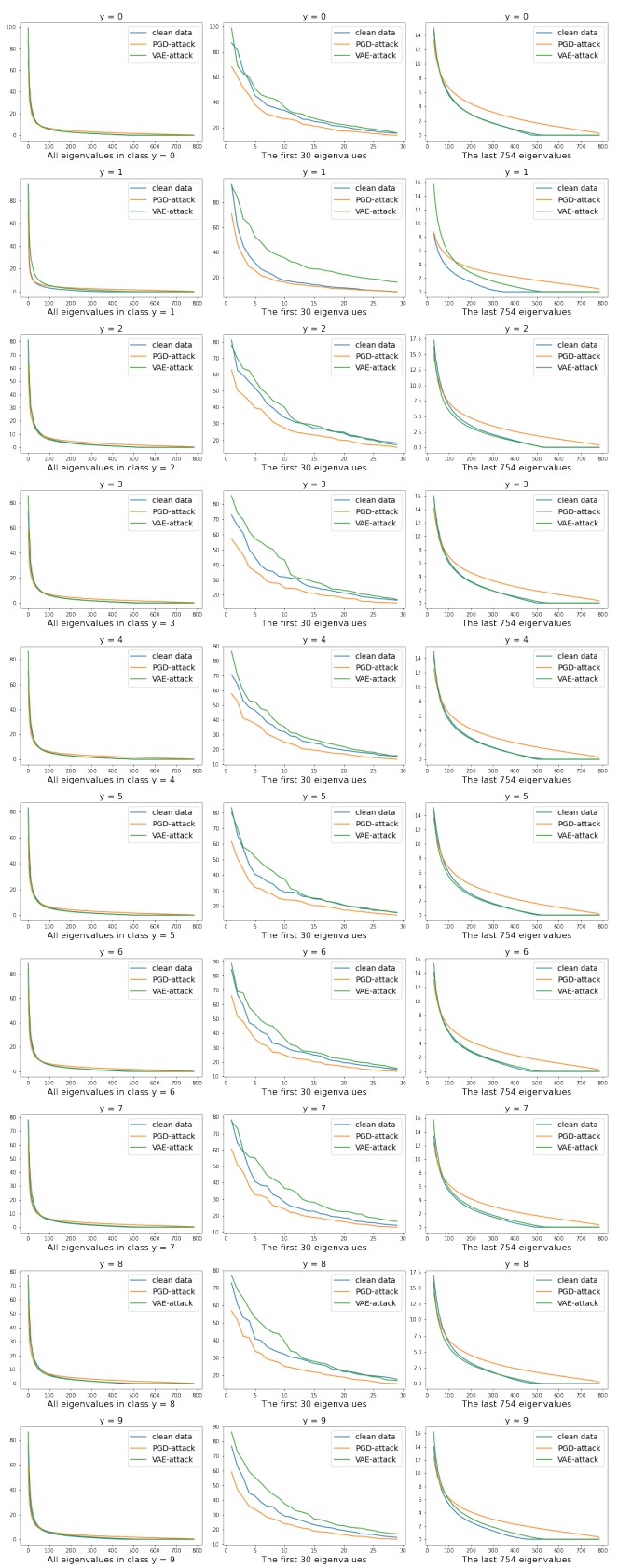

Figure 3: Eigenvalues of the covariance matrices of MNIST for all the 10 classes

**Generative model can help robust training**

**Theorem 9** (Adversarial learned parameters). *The Adversarial learned features of problem 10 under finite samples is $\boldsymbol{\mu}_{ls} = \hat{\boldsymbol{\mu}}$ and $\boldsymbol{\Sigma}_{ls} = \hat{\boldsymbol{U}}\boldsymbol{\Lambda}^{ls}\hat{\boldsymbol{U}}^T$ where $\boldsymbol{\Lambda}^{ls}$ is of the same form in Theorem 5 except replacing $\lambda_i$ by $\hat{\lambda}_i$ under strategy 3.*

Proof: Considering strategy 3. In this case

$$\boldsymbol{W} = \hat{\boldsymbol{U}}(\hat{\boldsymbol{\Lambda}_q} - \sigma^2)^{1/2}.$$

and

$$x' \sim (\hat{\boldsymbol{\mu}}, \boldsymbol{W}\boldsymbol{W}^T + \sigma^2 I).$$

Since

$$\boldsymbol{W}\boldsymbol{W}^T + \sigma^2 I = \hat{\boldsymbol{U}} \begin{bmatrix} \hat{\boldsymbol{\Lambda}_q} & 0 \\ 0 & \sigma^2 I \end{bmatrix} \hat{\boldsymbol{U}}^T.$$

In words, the eigenvectors of the dsitribution $\hat{\boldsymbol{U}}$ is the same as the one in perturbation $\boldsymbol{W}\Delta z$. The optimization problem is the same as the one we use in the proof of Theorem 5 if we replace $\lambda_i$ by $\hat{\lambda}_i$. $\quad\square$

If the data lie in a $q$ dimensional subspace, we do not neet to make the assumption on strategy 3.

