# OpenReview forum: "Disentangling Adversarial Robustness in Directions of the Data Manifold"
_ICLR.cc/2021/Conference — Reject_

### Official Review · AnonReviewer4 · 2020-10-20
**Disentangling Adversarial Robustness in Directions of the Data Manifold**

**Rating:** 6
**Confidence:** 5

**Review:**

The present paper proposes an interesting study about two different kinds of adversarials, depending on the area of the variance and manifold that is attacked. Although the results are promising, there is some lack of further experimentation to provide more meaningful results, so I would recommend providing more experimentation with more different and recent attacks and datasets to enhance the contribution. Hereby I provide some suggestions and minor comments per section, as well as some general
1.	Introduction Why the method is not applied experimentally to real datasets instead of just claiming that theoretically proved that it works? More details and experimentation should be provided about the application in these scenarios. Minor:
•	“training methods that use” (not using)
2.	Related work In general, the section should be extended with more details in the concepts: white/black box approaches, optimization methods (difference between first and zeroth). Missing reference for zeroth order optimization. Also, more recent attacks in the white box setting are available now, which should be preferred rather than “first generation” 2016/2017 attacks. Same for black box approach. Consider for example SL1D, HopSkipJump etc Stating that adversarial training is the only effecting defense is a big claim. Consider supporting this kind of discussions with more references. For example: Tramer, F., Carlini, N., Brendel, W., & Madry, A. (2020). On adaptive attacks to adversarial example defenses. arXiv preprint arXiv:2002.08347.
3.	Problem description Why do not use Carlini and Wagner attack (for example), rather than FGSM, as the former is a more powerful attack and more widely relevant in the state of the art. In line with previous comments, more recent approaches for this kind of attacks should be preferred.
4.	Theoretical analysis Minor:
•	To simplify (not “simply”) in 4.3 (excess risk analysis)
5.	Experiments The number of samples used for experimentation in each dataset should be provided in the main text, along with more details on the training process (apart from the appendix) Why do labels in Figure 2 not match clear names for data representation? (no need to keep track of the renaming from PGD to “norm-base attack”, just make an effort to keep naming consistent) Minor:
•	MNIST dataset should be capitalized (not “Mnist”)
•	“resnet” network should also be spelled as its proper acronym: ResNet (for example, as derived from Residual Networks)
•	“lies in a low dimension affine plane in R784. After” (capitalization after full stop)
6.	Conclusion The final results in Table 1 do not show a clear improvement with both, were in some cases the accuracy is reduced with respect to using a single method. For example, 89.50% with both vs 95.51 % when using only PGD-adv. This happens in the majority of cases. For this reason, a more detailed study on the contribution of each method to the defence should be provided, such as statistical tests and an extensive ablation study. The conclusion paragraph itself is too brief, just a brief summary rushing to the end. More discussion should be provided, along with more insights on the potential applications (for example, how to take advantage from this knowledge to design a new defence technique). General:
•	More references regarding the adversarial robustness trade-off should be added to the text and discussed, as a key point presented in the introduction and suggested as future work in the conclusion. Consider for example the following two:
1.	Su, D., Zhang, H., Chen, H., Yi, J., Chen, P. Y., & Gao, Y. (2018). Is Robustness the Cost of Accuracy?--A Comprehensive Study on the Robustness of 18 Deep Image Classification Models. In Proceedings of the European Conference on Computer Vision (ECCV) (pp. 631-648).
2.	Deniz, O., Pedraza, A., Vallez, N., Salido, J., & Bueno, G. (2020). Robustness to adversarial examples can be improved with overfitting. International Journal of Machine Learning and Cybernetics, 1-10.
•	English spelling should be revised thoroughly, since typos are found frequently.

---

> ### Author Response · Authors · 2020-11-17
> **Question Answering**
>
> We thank the reviewer for the comments and suggestions. We appreciate the time you spent on the paper and we apologize for typos and grammar mistakes. Below we address the concerns and comments that you have provided.
>
> Suggestions 1, 2, 4, 5, and 6:
>
> A: Thanks for your suggestion, they are very helpful for us to revise our paper. We revise each of the sections based on the suggestions. Specifically, we give more details and discussion about the robustness trade-off in section 5.3. We discuss the future works in detail in section 6.
>
> Question 3, why do you use FGSM but not a more powerful C&W attack?
>
> A: Thanks for the question. We choose FGSM-attack to show that latent space adversarial training does not work well in a simple FGSM-attack, and vice versa. In the updated paper, we have more details in this part.
>
> Concern 5, There is a robustness trade-off between regular adversarial examples and generative adversarial examples.
>
> A: Thanks for the comment. In the updated paper, we discuss the robustness trade-off in section 5.3. On MNIST, the trade-off between regular adversarial examples and generative adversarial examples is unavoidable. But the conflict between them is much less than the conflict between two norm-based attacks. I think the reason behind this robustness trade-off is that there is an overlap between the directions of the $q$ largest variance and the directions of small variance. Regular and generative attacks conflict with each other when they focus on the overlap directions. Please see the updated paper for more experiments and discussion.

---

### Official Review · AnonReviewer1 · 2020-10-26
**A well-motivated paper, but still needs more explanation regarding its main theoretical claims.**

**Rating:** 5
**Confidence:** 4

**Review:**

This paper investigates the differences between attack and defense in adversarial machine learning, when a) the adversarial perturbation is applied directly to the data samples, and b) when perturbation is applied to the latent space of a VAE or GAN which then generates the data. The above-mentioned investigation includes both theoretical and experimental analysis which are given in Section 4 and 5 of the paper, respectively.

Paper is well-motivated and is focused on some recent and interesting aspects of adversarial robustness. However, the theoretical treatment given in this paper only considers a very simple problem setting instead of providing a more comprehensive framework for future researches. Also, the paper's main claim may not be firmly established yet, and needs further mathematical work and clarification to become ICLR-ready. To be more specific, I have the following two main concerns about this paper.

#1:
Authors, both in the abstract and also several other places in the Introduction section, have made a number of huge claims about (theoretically) analyzing the fundamental differences of on-manifold and off-manifold attacks, in general. However, the theoretical analysis in this paper is completely centered around a highly-restricted linear generative model with a carefully-chosen design. Data generation model is also assumed to be a simple Gaussian model with infinitely many observed samples, which gives easy-to-handle analytic closed forms for the solution of all the optimization problems that one may encounter in this work. Therefore, the big claims made throughout the abstract and introduction section need to be significantly relaxed.

Also, it might not be a bad idea to re-organize the overall structure of the paper: Beginning with some experimentation and then showing the theoretical results as a simple case study. Otherwise, the limitations and also the simplifying assumptions behind presented theoretical analysis should be clearly mentioned both in abstract and Introduction.

#2:
I am not completely sure about the mathematical validity of the main claim of the paper (at least, from a theoretical standpoint), which is the "The comparison between Corollary 3 and Theorem 4". First, in my opinion, both Corollary 3 and Theorem 4 have shown that the true dimension underlying the data (or at least, the number of dimensions that the adversary is allowed to use for its attacks) appears in the excess risk bound, i.e.
$$
\mathcal{L}\_r\left(\Theta_{\star},\mathcal{D}\right)-\mathcal{L}\left(\Theta_{\star},\mathcal{D}\right)
\leq
\mathcal{O}\left(d\left(\lambda_{\min}L\right)^{-2}\right)
$$
which shows a linear dependence on $d$, and
$$
\\mathcal{L}\_{\\mathrm{ls}}\\left(\\Theta\_{\\star},\\mathcal{D}\\right)
-\\mathcal{L}\\left(\\Theta\_{\\star},\\mathcal{D}\\right)
\\leq
\\mathcal{O}\\left(qL^{-2}\\right).
$$
which shows a similar increasing behavior w.r.t. $q$. However, authors have preferred to only use the lower-bound of Theorem 4 which does not contain the factor $d$. Why? that should be properly explained. Moreover, the fact that the upper bound of Corollary 3 scales with $q$ while that of Theorem 4 scales with $d$ makes sense, since in the latent space adversarial training, adversary is restricted to use only $q\\leq d$ possible directions to form its attack.

The other crucial difference between the results of Theorem 4 and Corollary 3, is the appearance of $\\lambda_{\\min}$ in Theorem 4, but not Corollary 3. Authors have concluded that this difference must have something to do with the fundamental differences between on-manifold and off-manifold attacks. But, that is mainly due to the crucial difference in the mathematical meaning of Lagrange multiplier $L$  in the two formulations (Eq. (6) and Eq. (4)). In one of them, we have $\\Vert\\Delta\\boldsymbol{x}\\Vert\\leq\\epsilon$, where the perturbation is $\\Delta\\boldsymbol{x}$ itself. But in the other formulation we have $\\Vert\\Delta\\boldsymbol{z}\\Vert\\leq\\epsilon$, while the perturbation is $\\boldsymbol{W}\\Delta\\boldsymbol{z}$. So, for a fixed perturbation intensity $\\epsilon$, the magnitude of Lagrange multiplier $L$ needs to be adjusted between the two formulation (w.r.t. to spectrum of $\\boldsymbol{W}$). My guess is that it would justify the appearance of $\\lambda_{\\min}$ in Theorem 4 but not in Corollary 3. I'd like to know the authors' response on this issue as well.

There are several typos or grammatical errors in the paper. Some are listed below:

page 2: The attacker have -> should be corrected.
page 2: argumented -> augmented.
page2: VAE is also be used to train robust model -> should be rephrased.

page 4: Formulation of $\\boldsymbol{W}_{\\mathrm{ML}}$ may not be correct. The matrix on the r.h.s. might not have a real square root.
page 4: Data -> data.

page 5: simply -> simplify.

page 6: $q$-dimensional

At this stage, I cannot recommend this paper for publication at ICLR 2021 unless the above issues are answered properly.

---

> ### Author Response · Authors · 2020-11-17
> **Question Answering**
>
> We thank the reviewer for the comments and suggestions. We appreciate the time you spent on the paper and we apologize for typos and grammar mistakes. Below we address the concerns and comments that you have provided.
>
> Concern 1(a): The abstract and introduction overstate the theoretical results.
>
> A: Thanks for the comment. We carefully checked and revised the abstract and introduction. In the updated paper, we emphasize that the theoretical results are done in Gaussian mixture model case with linear generator. If it is still misleading please let us know.
>
> Suggestion 1(b): Beginning with some experimentation and then showing the theoretical results.
>
> A: Thanks for the suggestion. I think this is a good idea. But many experiments are done based on the theory. At this stage, we keep the current structure of the paper. We will consider how to reorganize the presentation of the paper.
>
> Concern 2(a): Prefer to only use the lower bound.
>
> A: Thanks for the question. We emphasize the lower bound because we want to state that \lambda_min also exists in the lower bound. In the updated paper, we discuss both the lower and the upper bound.
>
> Concern 2(b): The comparison of the bounds in Corollary 3 and Theorem 4.
>
> A: Thanks for the question. Your concern that L should be different in Corollary 3 and Theorem 4 is right. We realize that using the same letter L in Corollary 3 and Theorem 4 is misleading. Firstly, the perturbation intensity \epsilon in original space attack and latent space attack are on different scales. Secondly, the perturbations \Delta x and W \Delta z are different. Hence, the corresponding Lagrange multiplier should be different in Corollary 3 and Theorem 4. We use L’ in Theorem 4 in the updated paper.
>
> We think your main concern is that the eigenvalues may be absorbed in L’ in Theorem 4. So it is unfair to compare the bounds in Corollary 3 and Theorem 4.
>
> Our answer is that Corollary 3 and Theorem 4 are correct. \lambda_min exists in Corollary 3 but not in Theorem 4.
>
> Technically, one can show that the order of L and L’ only depends on \epsilon and \epsilon’. Intuitively, as it is demonstrated in Figure 1 (a), the perturbation constraint (the black block) misaligns with the shape of the data manifold (the ellipse). So the excess risk will carry the information of the shape of the manifold, which is the \lambda_min. Therefore, \lambda_min exists in O(d(\lambda_min L’)^-2) in Theorem 4. Similarly, as we demonstrate in Figure 1 (c), the perturbation constraint aligns with the data manifold. Hence, O(qL^-2) in Corollary 3 does not contain eigenvalues. We give more discussion on these two Theorems in the updated paper below the Theorem.
>
> Minor: The formula of W_ML may not be correct. The matrix on the r.h.s. might not have a real square root.
>
> A: The formula is correct. Firstly, it is directly copied from the original paper of P-PCA (Tipping et al., 1999). Secondly, the r.h.s have a real square root since \lambda_i-\sigma^2>=0 for i<=q. Each of the top q eigenvalues will always be larger than the average of the last d-q eigenvalues.

---

### Official Review · AnonReviewer3 · 2020-10-27
**Review for Disentangling Adversarial Robustness in Directions of the Data Manifold**

**Rating:** 4
**Confidence:** 3

**Review:**

*Summary

This paper analytically considers two flavours of adversarial training in a Gaussian mixture model.  The first uses regular adversarial examples, and the second uses examples drawn from a generative model.  The authors show that the adversarial perturbations generated in the two cases differ in a cleanly-characterisable  way: in the first case the perturbations differ from real data in a direction aligned with the smallest eigenvalues  of the data covariance. In the latter case the perturbations are in a direction aligned with the largest eigenvalues.  Experimental results on MNIST and CIFAR are presented to illustrate how the analysis transfers to real datasets.

*Positives

The analytical results on the mixture-of-Gaussians setting are very interesting. It's nice how the authors are able to get algebraic results in this setting. As far as I am aware this is novel (although I am not an expert).

The direct comparison between two different forms of adversarial examples (on- and off-)manifold is a nice framing that illustrates the difference between the two types of adversarial training.

*Concerns

The paper seemed quite rushed, with a lot of spelling mistakes. Some of these are detailed in the final section.

The exposition of the theory could be motivated much better. As a representative example, in section 4.2 three alternative ways of generating the latent-space perturbations are presented, but there is no motivation describing how they are qualitatively different. Furthermore, it's not spelled out very clearly if the data itself is a lower-dimensional distribution embedded in a higher dimensional space (the usual setting for the 'manifold hypothesis' and proposed to be  a very important reason for adversarial examples). It seems like this is assumed in Corollary 3 (rank \Sigma_* = q) but then  not assumed in theorem 4, since then \lambda_min = 0.

Most crucially, I'm not sure that the experiments are very convincing that the phenomenon explored in the mixture-of-Gaussians setting is actually present in real data. Firstly, it's quite hard to make conclusions from figure 2 when the first and second eigenvalues are not shown. The authors argue that the dynamic range is too great to plot the values meaningfully, but a logarithmic y-axis would suffice. Looking at the first column, we can see that the norm-based attack has a higher value than the VAE attack, in the reverse of what the authors claim would be the expected behaviour. In the second set of experiments, the authors claim that training against the PGD attack does not transfer to defense against the VAE attack. Actually the accuracy against the VAE attack increases from 42% to 52%, which does seem to indicate transferability to me. The authors also claim that unlike adversarial training for defense to l_0, l_1, l_\infty attacks, the regular and generative examples have no robustness trade-off (i.e. you can train a model to be robust to both at once). However, the data seems to indicate that there is a trade-off. On MNIST, the PGD-trained model gets 95.51% accuracy on a PGD attack. The VAE-trained model gets 96.66% accuracy on the VAE attack. The jointly-trained model gets 89.5% accuracy on the PGD attack and 90.28% accuracy on the VAE attack, so it deteriorates in both cases. It's true there is perhaps less trade-off than expected, but the claim their method 'exhibit no robustness trade-off' seems unsupported by the evidence.


*Recommendation

Overall I recommend to reject this paper.
While the analytical results are nice, the experiments are not very convincing that the analysis carries into real data.
Furthermore, the paper as a whole seems rushed: missing details in the experimental section and a lack of motivation in the theory section.

*Questions

Can the authors explain the non-shown eigenvalues in the experiments, and discuss in more detail how their claim about the robustness trade-off is supported by the evidence?

*Minor points

I'm not an expert, but I think the description of PDG is an incorrect characterisation for norm other than the \ell_\infty norm. In particular, the step taken being a multiple of the sign of the gradient is only correct for the \ell_\infty norm, since there it corresponds to the steepest descent step. For e.g. the \ell_2 norm we should instead divide by the norm of the gradient, and so on.  In the experiments the 'both-adv' column is uniformly bold, even though in multiple rows it performs worse than the other models. For instance, the FGSM-adv model performs better under an FGSM-attack than the 'both-adv' model for CIFAR, so should be in bold (under the general rule that bold means the best-performing model unless otherwise specified).
The experimental results are quite sparse. In particular, I attempted to replicate the clean data results from figure 2 but was unable to with the details in the paper. Was there any normalization applied to the MNIST data before computing the covariance?

*Typos:

Section 3, para 1: '. Adversarial training is to solve' -> '. The goal of adversarial training is to solve'
Section 4.1, after equation 3: '. Adversarial training is to solve' -> '. The goal of adversarial training is to solve'
Theorem 6: 'The optimal solution of problem in' -> 'The optimal solution of the problem in '
Page 7, para 1: 'class 1 and 2' is described as class 1 and 2 and class 0 and 1.
Page 7, para 2: 'closed' -> 'close'
Page 8, para 3: 'Figure 1' -> 'Table 1'
Page 8, para 5: 'amplifying low variance of distribution' not grammatical sentence.
Page 8, para 6: 'we can defense all' -> 'we can defend all the attacks'

---

> ### Author Response · Authors · 2020-11-17
> **Question Answering**
>
> We thank the reviewer for the comments and suggestions. We appreciate the time you spent on the paper and we apologize for typos and grammar mistakes. Below we address the concerns and comments that you have provided.
>
> Concern 1: Motivation of the three ways of generating adversarial examples.
>
> A: Thanks for the suggestion, we should state the motivation clearly. When the encoder is a probabilistic model, choosing z with the highest probability, or sample it from the distribution are all reasonable. For the decoder, the reason is the same. So we specify different strategies. We think that they are the same because of the law of large numbers. In lemma 1, we show that they are equivalent and we don’t need to worry about the choices. We state the motivation in the revised paper.
>
> Concern 2: The assumptions on Corollary 3 and Theorem 4 is different.
>
> A: Thanks for the comment. Theorem 4 includes the general case and the low dimensional case, which is the assumption on Corollary 3. When \lambda_min=0, the excess risk = +\infty. Intuitively speaking, when the adversarial example x’ moves outside the data distribution, we have p(x’)=0. Then the negative log-likelihood loss function, -log p(x’), equals to +\infty. We emphasize this situation and give more discussion in the updated paper.
>
> Concern 3: The eigenvalues shown in Figure 2 are inconsistent with the theory.
>
> A: Thanks for your careful checking on this experiment. We check the experiments and find that we make a small mistake using the EVD library. We show the new results in the updated paper in Figure 2 without missing any eigenvalues. The results are consistent with the theory. We show the other 8 classes in Appendix B.3, Figure 3. Now we think you can replicate the experiments.
>
> Concern 4: Explanation of the experiment that PGD-Adv increases the test accuracy from 42% to 52% on VAE-attack.
>
> A: Thanks for the comment. In the updated paper, we discuss the benefits of original space adversarial training on VAE-attack. As indicated in Theorem 6 and the experiments shown in Figure 2, original space adversarial training will amplify the small eigenvalues in the first q dimension. But it fails to amplify the large eigenvalues in the first q dimension. So PGD-adv can increase the performance on VAE-attack but not good enough.
>
> Concern 5: There is a robustness trade-off between regular adversarial examples and generative adversarial examples.
>
> A: Thanks for the comment. The claim ‘exhibit no robustness trade-off’ is based on the results on CIFAR-10. This is a typo and it is misleading. In the updated paper, we discuss the robustness trade-off in section 5.3. On MNIST, the trade-off between regular adversarial examples and generative adversarial examples is unavoidable. But the conflict between them is much less than the conflict between two norm-based attacks. I think the reason behind this robustness trade-off is that there is an overlap between the directions of the $q$ largest variance and the directions of small variance. Regular and generative attacks conflict with each other when they focus on the overlap directions. Please see the updated paper for more experiments and discussion.
>
> Minor 1: Formula of PGD.
>
> A, Yes, the formula of PGD is for l_\infty norm attack. We replace it with a more general version.
>
> Minor 2: The number in bold.
>
> Thanks for pointing it out and we fixed the number in bold in Table 1.

---

> > ### Comment · AnonReviewer3 · 2020-11-23
> > **Reply**
> >
> > Thank you for the reply, and the substantially updated manuscript.
> > The extra motivation for the sampling of adversarial examples is welcome. However I still find it slightly hard to follow the main theoretical portion of the paper. Unfortunately I don't have a precise statement of what could be improved, but I think a clear diagram marked with the important quantities $W$, $\Lambda$, $\Sigma$ etc would make the whole section much easier to follow. As it is, figure 1 doesn't add very much to the paper.
> >
> > Secondly, I still feel that the experimental section doesn't make a strong argument of the transferrability of the theoretical analysis to a real-world setting. There's also a multitude of typos in the new sections and I am a bit worried about the robustness/correctness of your results given that you could find a mistake in the MNIST experiments so easily.
> >
> > Overall, I think the idea and theoretical analysis is very interesting, and the experimental results are certainly encouraging. With a bit more work, and a couple of revisions of the arguments and experiments I think it's likely this paper could be accepted. But at the moment it's not quite there.

---

### Official Review · AnonReviewer2 · 2020-10-29
**The paper mainly shows that adversarial robustness can be decomposed into small variance directions and large variance directions of the data manifold.**

**Rating:** 6
**Confidence:** 3

**Review:**

In this paper, the author mainly show that adversarial robustness can be disentangled in small variance directions and large variance directions.. Theoretically, they also investigated the excess risk and optimal saddle point of the minimax problem of latent space adversarial training.

Positive:
1.  found the regular adversarial examples attack tend to lies in small variance directions of the data.
2. found generative adversarial examples attack towards to the large variance directions of the data.
3. explore standard adversarial training as well as latent space adversarial training to deal with  on-manifold and off-mainfold issue.
4. The theory analysis may be useful to use original/latent adversarial training to increase the model robustness.


Negative:
1. The theoretical analysis is mainly based in probabilistic principle component analysis, a linear generative model. The extension to nonlinear model is unclear, which may be more common in practice.
2. In addition to LeNet and ResNet, it may be more convincing to test two more extra models to confirm the theoretical findings. The analysis rely on the eigenvalues, what if the original features are in high-dimensional space. computing eigenvalue decomposition may be expensive.
3. In the Table 1, it seems that using both regular adversarial examples and generative adversarial examples sometime does not obtain test accuracy, any more discussion?

In summary, the authors provide a theoretical study on theoretical analysis of the attacking mechanisms of the two kinds of adversarial examples: regular and generative adversarial examples. They should w that adversarial robustness can be disentangled in directions of the data manifold.  Such finds may be useful in designing defense algorithms.

---

> ### Author Response · Authors · 2020-11-17
> **Question Answering**
>
> We thank the reviewer for the comments and suggestions. We appreciate the time you spent on the paper and we apologize for typos and grammar mistakes. Below we address the concerns and comments that you have provided.
>
> Comment 1: The extension to nonlinear models is unclear.
>
> A: Yes, we agree. It is unclear how to analyze the nonlinear model. Technically speaking, the difficulty is that we cannot convert an optimization problem with nonlinear models to a convex problem. But for linear models, we may find a way to convert the target problem to convex problems. Currently, we can only provide experiments on nonlinear generative models. Since this is an open problem, we add it in future works (Page 9) in the updated paper.
>
> Comment 2(a): Maybe more convincing to test two more extra models to confirm the theoretical findings.
>
> A: Thanks for the suggestion. We will do more experiments on different models. Because many reviewers besides you concern more about the results for the jointly-train model, we provide more experiments on the comparison of robustness trade-off (section 5.3) at this stage.
>
> Comment 2(b): Computing eigenvalue decomposition may be expensive.
>
> A: Yes, this is an issue if we want to use this property to craft adversarial examples without access to the target model. In our paper, we did not discuss the computational cost because we only use EVD for analysis. In future works, when we design defense algorithms based on EVD, we should carefully compare the computational cost of EVD and the benefits that these adversarial examples bring to us.
>
> Comment 3: Using both regular adversarial examples and generative adversarial examples do not obtain test accuracy.
>
> A: Yes, on MNIST, there is a small conflict between regular adversarial examples and generative adversarial examples. In the updated paper, we provide more discussion and experiments in section 5.3: robustness trade-off. We see that there is an overlap between the directions of the $q$ largest variance and the directions of small variance. We think this is a possible reason for the robustness trade-off between regular and generative attacks. Please see our updated paper.

---

### Author Response · Authors · 2020-11-24
**General Response: Revision Updated**

We thank the reviewers for the comments and suggestions. We appreciate the time you spent on the paper and we apologize for typos and grammar mistakes. We tried our best to find and fix them. Below we summarize the revision.

Abstract and Introduction: We revised the claim of theoretical results.

Related works: We had more discussion on related works.

Problem description: We changed the formula of PGD attack.

Theoretical analysis: We gave more discussion about the results of Corollary 3 and Theorem 4.

Experiments:

1, We fixed the first experiment about the comparison of eigenvalues.

2, We discussed the test accuracy of PGD-adv on latent space attack.

3, We gave more discussion on the robustness trade-off between the generative and the regular attack in Sec 5.3.

Conclusion: We gave more discussion on future works.

---

### Decision · Program_Chairs · 2021-01-07
**Final Decision**

**Decision:**

Reject

**Comment:**

In this paper, the authors theoretically analyzed the attacking mechanisms of the two kinds of adversarial examples in the Gaussian mixture data model case and proved that adversarial robustness can be disentangled in directions of the data manifold. The reviewers commonly felt that the idea and theoretical analysis in this paper are interesting, but experiments are not satisfactory.

At the current status, they still have a main concern regarding the correctness of comparison between the results of Theorem 4 and Corollary 3 (which is the heart of their theoretical claims, the main message of the paper and the main motivation for experiments).

As a whole, this paper has some merits but the authors still cannot clarify some concerns raised by some reviewers.